# The influence of evidence volatility on choice, reaction time and confidence in a perceptual decision

Ariel Zylberberg[1,2]*, Christopher R Fetsch[1,2], Michael N Shadlen[1,2]*

[1]Kavli Institute, Department of Neuroscience, Howard Hughes Medical Institute, Columbia University, New York, United States; [2]Zuckerman Mind Brain Behavior Institute, Department of Neuroscience, Howard Hughes Medical Institute, Columbia University, New York, United States

**Abstract** Many decisions are thought to arise via the accumulation of noisy evidence to a threshold or bound. In perception, the mechanism explains the effect of stimulus strength, characterized by signal-to-noise ratio, on decision speed, accuracy and confidence. It also makes intriguing predictions about the noise itself. An increase in noise should lead to faster decisions, reduced accuracy and, paradoxically, higher confidence. To test these predictions, we introduce a novel sensory manipulation that mimics the addition of unbiased noise to motion-selective regions of visual cortex, which we verified with neuronal recordings from macaque areas MT/MST. For both humans and monkeys, increasing the noise induced faster decisions and greater confidence over a range of stimuli for which accuracy was minimally impaired. The magnitude of the effects was in agreement with predictions of a bounded evidence accumulation model.

*For correspondence: ariel.
zylberberg@gmail.com (AZ);
shadlen@columbia.edu (MNS)

## Introduction

Decisions that combine information from different sources or across time are of special interest to neuroscience because they serve as a model of cognitive function. These decisions are not hard wired or reflexive, yet they are experimentally tractable. Psychologists have long sought to understand how the process of decision formation gives rise to three key observables (*Cartwright and Festinger, 1943*; *Audley, 1960*; *Vickers, 1979*). First there is the choice itself (left or right, coffee or tea), which determines accuracy in cases where a correct alternative can be defined. Second, there is the time it takes to reach a decision, which determines reaction-time (RT). RT furnishes a powerful constraint on models of decision-making, and is a defining element of the trade-off between speed and accuracy that characterizes most decisions. Third, decisions are often accompanied by a graded degree of belief in the accuracy or appropriateness of the choice. This belief, referred to as decision confidence, influences many aspects of behavior: how we learn from our mistakes, plan subsequent decisions, and communicate our decisions to others. A model of the decision process ought to explain not just choices but all three of these observables in a quantitative fashion.

The family of bounded evidence accumulation models, including drift diffusion, race and attractor models, offers one such framework for linking choice, reaction time and confidence [for reviews, see *Gold and Shadlen (2007)*; *Shadlen and Kiani (2013)*]. These models depict the decision process as a race between competing accumulators, each of which integrates momentary evidence for one alternative and against the others. The decision terminates when the accumulated evidence for one alternative, termed a decision variable (DV), reaches a threshold or bound, thereby determining both the choice and the decision time. Confidence in the decision derives from a mapping between the DV and the probability that a decision based on this DV will be correct. The mapping is thought

**eLife digest** Many of our decisions are made on the basis of imperfect or 'noisy' information. A longstanding goal in neuroscience is to work out how such noise affects three aspects of decision-making: the accuracy (or appropriateness) of a choice, the speed at which the choice is made, and the decision-maker's confidence that they have chosen correctly.

One theory of decision-making is that the brain simultaneously accumulates evidence for each of the options it is considering, until one option exceeds a threshold and is declared the 'winner'. This theory is known as *bounded evidence accumulation*. It predicts that increasing the noisiness of the available information decreases the accuracy of decisions made in response. Counterintuitively, it also predicts that such an increase in noise speeds up decision-making and increases confidence levels.

Zylberberg et al. have now tested these predictions experimentally by getting human volunteers and monkeys to perform a series of trials where they had to decide whether a set of randomly moving dots moved to the left or to the right overall. Using a newly developed method, the noisiness of the dot motion could be changed between trials. The effectiveness of this technique was confirmed by recording the activity of neurons in the region of the monkey brain that processes visual motion information.

After each trial, the humans rated their confidence in their decision. By comparison, the monkeys could indicate that they were not confident in a decision by opting for a guaranteed small reward on certain trials (instead of the larger reward they received when they correctly indicated the direction of motion of the dots).

In both humans and monkeys, increasing the noisiness associated with the movement of the dots led to faster and more confident decision-making, just as the bounded evidence accumulation framework predicts. Furthermore, the results presented by Zylberberg et al. suggest that the brain does not always gauge how reliable evidence is in order to fine-tune decisions.

Now that the role of noise in decision-making is better understood, future experiments could attempt to reveal how artificial manipulations of the brain contribute both information and noise to a decision. Other experiments might ascertain when the brain can learn that noisy information should invite slower, more cautious decisions.

to incorporate the decision time or the state of the competing (losing) accumulator(s), or both (*Vickers, 1979*; *Kiani and Shadlen, 2009*; *Zylberberg et al., 2012*; *Kiani et al., 2014*; *Van den Berg et al., 2016*). The noisiness of the momentary evidence causes the DV to wander from its starting point, as in Brownian motion or diffusion, whereas the expectation (i.e., mean) of the momentary evidence increments or decrements the DV deterministically. Noise is the main determinant of both RT and confidence when signal-to-noise is low, that is when choices are more stochastic (less accurate). Recent evidence from neurophysiology (*Kiani and Shadlen, 2009*), brain stimulation (*Fetsch et al., 2014*), and psychophysics (*Kiani et al., 2014*) supports such a mechanism.

If the bounded accumulation of noisy evidence underlies choice accuracy, RT and confidence, then a selective manipulation of the noise should produce quantitatively consistent effects on all three measures. Specifically, were it possible to leave unchanged the expectation of each sample of momentary evidence while boosting the noise associated with it, then the bounded accumulation of the noisier samples should lead to (i) lower accuracy when the expectation of the momentary evidence is strong, (ii) faster reaction times when the momentary evidence is weak, and (iii) increased confidence when the momentary evidence is weak. The basic insight behind the latter two predictions is that with greater volatility, the DV tends to diffuse more quickly away from the starting point to achieve levels nearer the termination bound which are ordinarily associated with stronger evidence and thus greater confidence (*Figure 1*).

These predictions have not been tested thoroughly, because a controlled method for selectively increasing noise is not known. A dissociation between accuracy and confidence led *Rahnev et al. (2012)* to conclude that transcranial magnetic stimulation (TMS) increased the neural noise associated with the representation of a visual pattern, and a similar dissociation

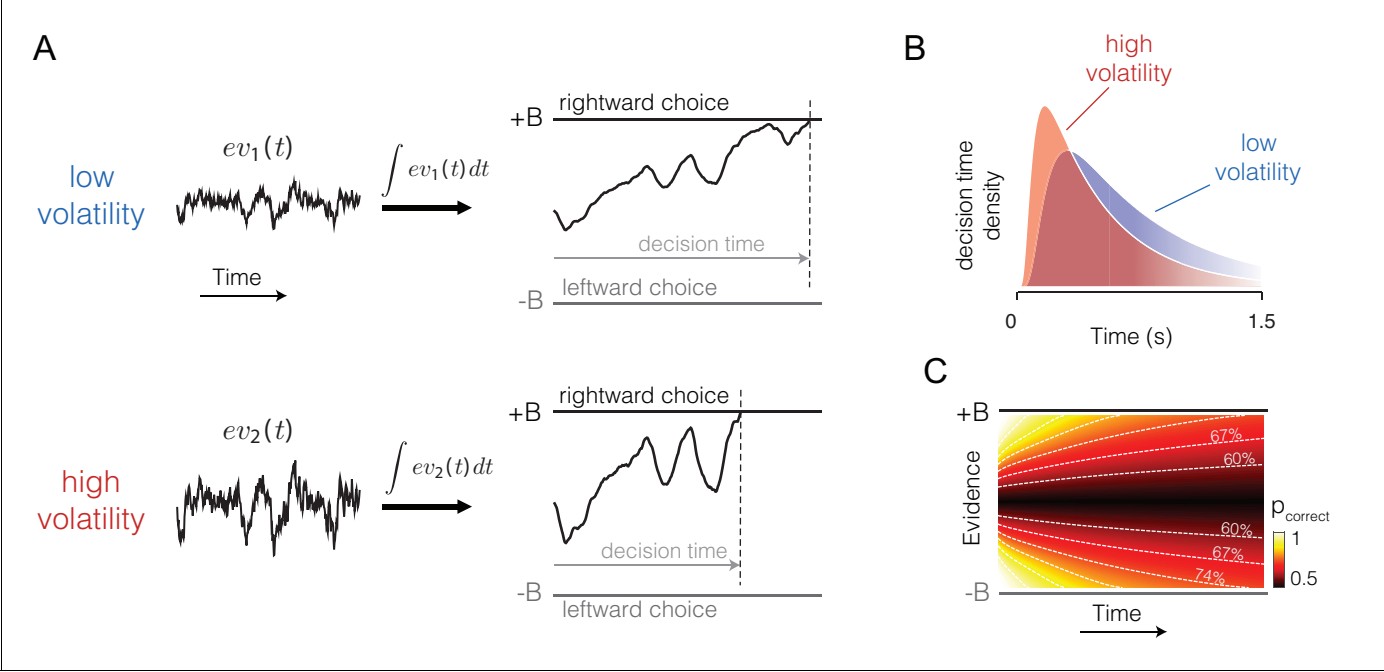

**Figure 1.** Predicted influence of volatility on reaction time and confidence under bounded evidence accumulation. (**A**) $ev_1(t)$ and $ev_2(t)$ represent the time course of momentary evidence, for stimuli of low and high volatility, respectively. In bounded accumulation models, momentary evidence is integrated over time, until the accumulated evidence (decision variable, DV) crosses one of two bounds, here at $\pm$ B. Bound-crossing simultaneously resolves the choice that is made and the time it takes to make it (decision time; the reaction time also includes a nondecision component, not shown). With greater noise, the decision variable tends to diffuse more rapidly, leading to faster responses. (**B**) Illustration of the effect of volatility on the distribution of decision times, for two bounded accumulation models that have the same drifts and bound heights but different diffusion coefficients. As in the single trial example, higher variance leads to faster responses. (**C**) Heat map depicting the association between the state of accumulated evidence and the probability that a decision rendered on this evidence is correct. The structure in this graph arises because there are several difficulty levels. More reliable stimuli (e.g., high motion coherence), which support high accuracy, contribute to large vertical excursions of the decision variable away from the starting point (midpoint of the ordinate) at short elapsed time, whereas less reliable stimuli contribute to equivalent vertical excursions at later times. Example probability contours are depicted with dashed lines. Because the volatility of the stimulus is not explicitly represented in this map, higher volatility would lead to greater confidence, because the decision variable diffuses more quickly from the starting point, leading paradoxically to states that are normally associated with more reliable sources.

led *Fetsch et al. (2014)* to conclude that cortical microstimulation (μStim) might affect both the mean and the variance of the representation of motion by neurons in the extrastriate visual cortex (areas MT/MST). However, characterization of these effects of TMS and μStim was inferred from behavior. Similarly, psychophysical studies that attempted to increase the noise through changes in the visual stimulus (*de Gardelle and Summerfield, 2011*; *Zylberberg et al., 2014*; *de Gardelle and Mamassian, 2015*) or attentional state (*Rahnev et al., 2011*; *Morales et al., 2015*) did not characterize the influence of these manipulations on the neural signals that the brain accumulates to form a decision.

We therefore sought a method to manipulate the variance associated with the neural representation of momentary evidence without affecting its mean. We achieved this with a manipulation of the motion information in a random dot motion (RDM) display, by adding a second level of randomness which increased its volatility but was unbiased with respect to the strength and direction of motion evidence. We verified that the manipulation has the desired properties by recording from direction selective neurons in the middle temporal (MT) and medial superior temporal (MST) areas of the macaque visual cortex. Neurons in these areas are known to represent the momentary evidence in tasks identical to those in our study (*Salzman et al., 1990*; *Celebrini and Newsome, 1995*; *Ditterich et al., 2003*; *Fetsch et al., 2014*). We then used the volatility manipulation to test the influence of noise on the three observables of choice behavior—accuracy, RT and confidence—in monkeys and humans.

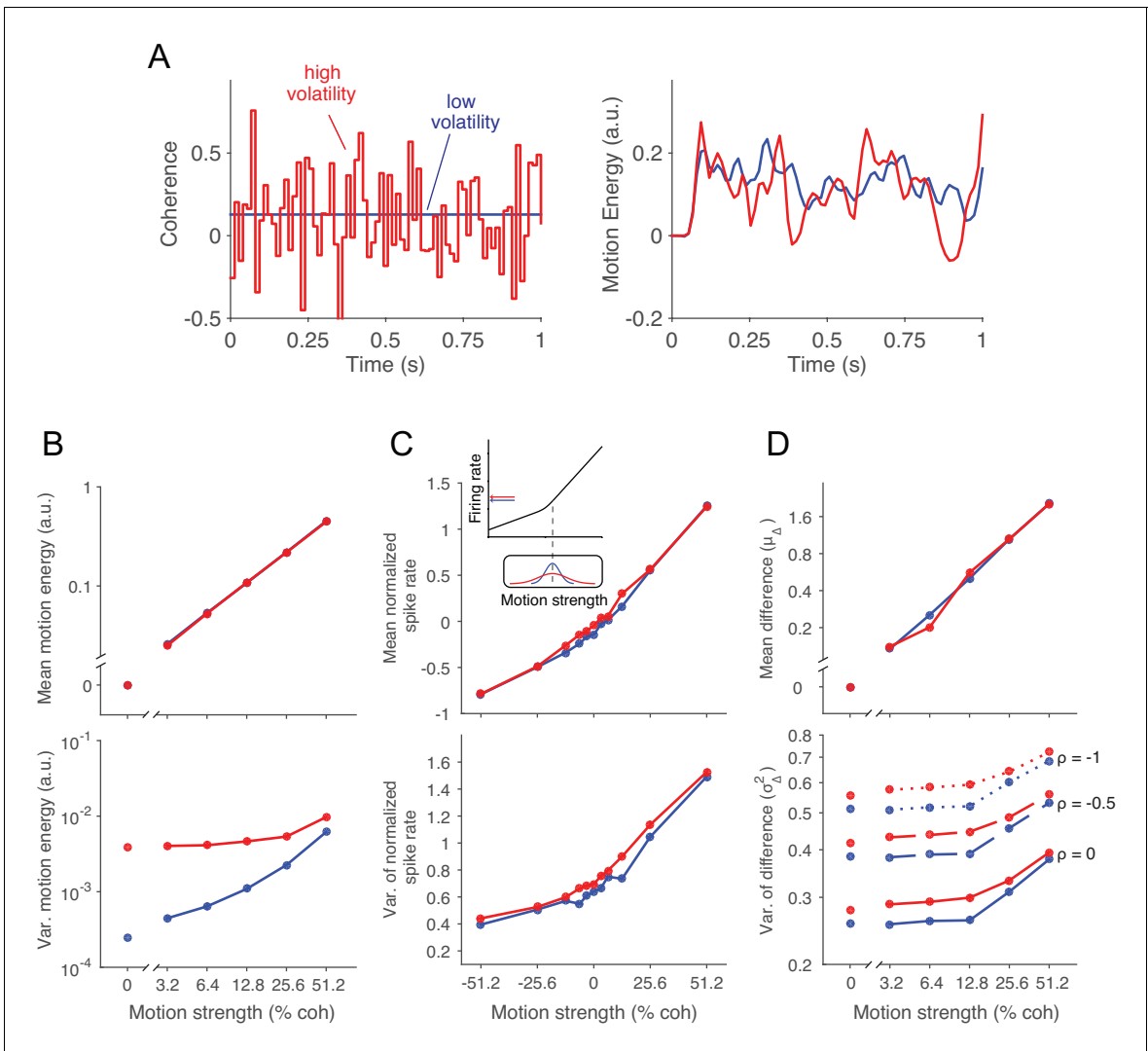

**Figure 2.** Doubly stochastic random dot motion selectively influences the variance of evidence. Low and high volatility conditions are indicated by blue and red, respectively, in all panels. (A) Motion strength as a function of time for example stimuli with low and high volatility. Left: the coherence parameter is 0.128 for both stimuli, but in the high volatility condition this is the mean of a Gaussian distribution (S.D. = 0.256) that is sampled on every video frame (75 Hz). Right: motion energy in favor of the positive direction. Both volatility conditions yield variation in the motion information, but the red curve exhibits more variation. (B) Mean and variance of the motion energy in support of the true direction of motion, computed separately for trials of low and high volatility (N = 66,805 trials). For all motion strengths, the mean (upper) is not affected by the volatility manipulation, whereas the variance is larger in the high volatility condition. Note the log scale for both axes. (C) Mean and variance of the neuronal response from direction selective neurons in areas MT/MST (N = 26 single units and 21 multiunit sites; see Materials and methods). Spike counts were obtained from a 100 ms window beginning 100 ms after stimulus onset and standardized (z-score) for each neuron or multiunit site. The volatility manipulation produced a small increase in the average firing rate at the low coherences (upper). This increase is likely due to the rectification of the noise by the nonlinear response of the neuron to motion in the preferred and anti-preferred directions, as sketched in the inset. The variance parallels the mean, but volatility has a more marked effect on variance at weak motion strengths. Note the linear scales. (D) Mean and variance of a difference between opposing pools of neural signals. The graphs extrapolate from panel C by constructing two pools of 100 or more neurons sharing a common preferred or anti-preferred direction, respectively. The mean of the difference variable ($\mu_\Delta$) is similar for both volatility conditions (upper), whereas the variance of the difference variable ($\sigma_\Delta^2$) is greater under high volatility (lower). This relationship is shown for three values of correlation ($\rho$) between the pools which span the plausible range. The correlation is negative because the opposing pools respond oppositely to fluctuations in the motion stimulus.

The following source data and figure supplement are available for figure 2:

**Source data 1.** Mean and variance of the neuronal response and of a difference variable between pools of neurons with opposite direction preferences.

**Figure supplement 1.** Variance of the momentary evidence from model fits.

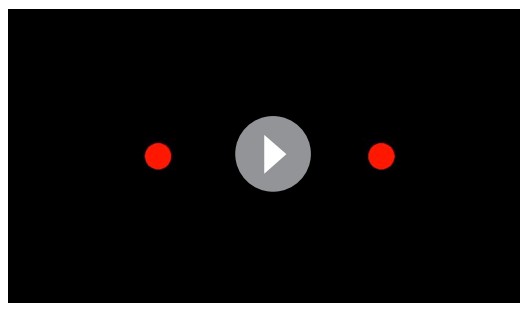

**Video 1.** Example motion stimuli. The movie shows the low and high volatility examples depicted in *Figure 2A*. For illustration purposes, before showing the moving dots we indicate the coherence, volatility and direction of motion. These were not displayed to the participants in the experiment.

## Results

### A manipulation that mimics the addition of noise to the visual cortex

The standard RDM stimulus is itself stochastic, meaning that a particular movie (e.g., shown on a trial) is an instantiation of a random process that conforms to an expected motion strength and direction. On each video frame, a dot that had appeared $\Delta t$ ms ago is either displaced (i.e., moved) or replaced by a new dot at a random location within the stimulus aperture. The determination of displacement versus replacement is in accordance with a flip of a biased coin, and the magnitude of this bias confers the motion strength, which we refer to as a motion coherence ($c$). The sign of $c$ indicates the direction of the displacement along an axis (e.g., up/down). Thus the probability of displacement (or unfairness of the coin) is |c|. The randomly replaced dots fall in the neighborhood of other dots (recently displayed) and thus contribute random motion in both directions. In the standard RDM, the coherence, $c$, is fixed for the duration of an experimental trial (e.g., $c = 0.13$; *Figure 2A*, left, blue line). Here we introduce a second layer of variability, wherein the mean of $c$ is fixed for the duration of a trial but the value of c varies randomly from video frame to video frame (*Figure 2A*, left, red line). We will refer to trials that employ this doubly stochastic RDM as the 'high volatility' condition and those that use the standard RDM as 'low volatility'.

This description explains how the stimulus is generated, but it does not explain what effect it should have on perception or on the neural processing of motion. The construction of the RDM we use is in video frames displayed every 1/75 of a second. The visual system blurs these images over time, leading for example to the illusion that many more dots are present simultaneously than are actually displayed. The right panel of *Figure 2A* applies an established motion filter (*Adelson and Bergen, 1985*) to the example movies parameterized by the low and high volatility traces shown in the left panel (see also *Video 1*). The filter extracts a time-blurred motion signal that provides a reasonable approximation to the firing rates of direction selective neurons in the primate visual cortex (*Britten et al., 1993*; *Rust et al., 2006*; *Hedges et al., 2011*). The example highlights the subtlety of the volatility manipulation by reminding us that the standard RDM is itself volatile (blue curve) such that the overall contour of both traces is similar. Nonetheless, the extra bumps and wiggles in the red trace result from the random variation in c.

A more systematic analysis of the motion energy, displayed in *Figure 2B*, reveals that the mean is identical for low and high volatility stimuli, for all motion strengths (upper panel), whereas the variance is larger for the high volatility stimuli (lower panel). The linear relationship between the mean motion energy and c is known (*Britten et al., 1993*), but the dependency of variance of the motion energy on c is less well characterized. For the low volatility condition (*Figure 2B* bottom, blue trace), the motion energy variance is dominated by the variance in the number of coherently displaced dots, which obeys a binomial distribution, hence the monotonic increase over the range of |c| = 0 to 0.5. For the high volatility condition (*Figure 2B* bottom, red trace), the overall increase in variance is not surprising, because we have added a second layer of variability. Note that the effect is strongest at the low coherences, where the distribution of c in the high volatility condition spans both positive and negative values.

These observations characterize the volatility present in the visual stimulus, but we are mainly interested in the noisy signals that the brain accumulates to form a decision. We therefore measured the impact of volatility on the response of direction selective neurons in cortical areas MT/MST (*Figure 2C*). These neurons represent the momentary evidence used by monkeys to guide their choice, reaction time and confidence (*Salzman et al., 1990*; *Celebrini and Newsome, 1995*; *Ditterich et al., 2003*; *Fetsch et al., 2014*) in motion discrimination tasks. As previously shown

(*Britten et al., 1993*), the firing rate of MT neurons increases linearly, on average, as a function of motion strength in the neuron's preferred direction ($c > 0$, *Figure 2C*, upper panel, blue trace). The firing rate decreases linearly, but less steeply, as a function of motion strength in the anti-preferred direction ($c < 0$), giving rise to a bilinear function. We refer to the shallower slope for $c < 0$ as rectification (*Britten et al., 1993*). These features are preserved under high volatility (red trace), but there is a subtle increase in firing rate at the low coherences, which is explained by the rectification of neural responses when the distribution of $c$ spans positive and negative values (*Figure 2C*, inset). The variance of the neural response is known to scale approximately linearly with firing rate (*Tolhurst et al., 1983*; *Vogels et al., 1989*; *Geisler and Albrecht, 1997*; *Shadlen and Newsome, 1998*). Thus the variance curves in *Figure 2C* (lower panel) parallel the means. The high volatility condition adds to the variance in a manner that is exaggerated at the low motion strengths, consistent with the motion energy analysis above.

We are now ready to consider the mean and variance of the quantity that is integrated toward a decision. We assume that the momentary evidence is the difference between the average firing rates from two pools of neurons with direction preferences for the two opposite directions (e.g., right-preferring minus left-preferring) (*Shadlen et al., 1996*; *Ditterich et al., 2003*; *Hanks et al., 2006*). The expectation of this signal can be estimated empirically by subtracting the mean firing rates of single neurons to motion in their preferred versus anti-preferred directions (*Figure 2D*). Notice that the rectification is now canceled by the subtraction.

The variance of this difference is more nuanced, drawing on two related considerations. First, because we did not record multiple single units simultaneously, we are not directly measuring the variance of the pools. Assuming a population of correlated neurons, the variance of the population mean differs from that of a single neuron by a multiplicative constant. For large pools, the variance is reduced to roughly $r\sigma^2$, where $r$ is the average pairwise spike-count correlation for neurons within the pool and $\sigma^2$ is the variance of the spike counts from a single neuron (see Materials and methods). In MT, $r$ is on the order of 0.2 for neurons with similar directional preferences (*Zohary et al., 1994*; *Bair et al., 2001*). An important implication of such correlation is that the beneficial effects of pooling saturate with modest number of neurons (e.g., 50–100; [*Zohary et al., 1994*; *Shadlen et al., 1996*]).

Second, the variance of the MT population comprises contributions from the variance in motion energy, described above, as well as a component that is independent of stimulus fluctuations. The opposing pool is assumed to share the component of variance originating in the stimulus, albeit of opposite sign, so the variances add rather than cancel in the difference. In contrast, the stimulus-independent component of shared variance (e.g., driven by fluctuations of arousal) should have the same sign in the two pools and thus cancel in the difference.

For a given coherence $c$ and volatility $v$, the variance of the difference in neuronal response between a pair of populations selective to the preferred and anti-preferred directions is given by:

$$\sigma^2_{\Delta|c,v} = r\left(\sigma^2_{c,v} + \sigma^2_{-c,v} - 2\rho\sigma_{c,v}\sigma_{-c,v}\right), \tag{1}$$

where $\sigma^2_{c,v}$ and $\sigma^2_{-c,v}$ are the variance of the spike counts for motion in the preferred and anti-preferred directions, $r$ is the average pairwise correlation for neurons within the same pool, and $\rho$ is the correlation between the two pools with opposite direction preferences. The variances on the right-hand side of *Equation 1* can be obtained from *Figure 2C*. However, without simultaneous recordings from neurons in the two pools, we cannot know how much of the variability is shared across neurons.

In *Figure 2D* we explored three different values of $\rho$: 0, $-0.5$ and $-1$ (with $r = 0.2$). Note that positive values of $\rho$ are unlikely because a large portion of the shared variability comes from stimulus fluctuations, which as stated above induce changes in firing rate of opposite sign in the two pools. Under the low volatility condition, the variance of the difference variable increases slightly as a function of motion strength. This is a consequence of rectification and the tendency for variance to parallel the mean firing rate. More importantly, the doubly stochastic stimuli lead to a marked increase in $\sigma^2_\Delta$, especially in the low coherence range where the impact on motion energy is greatest. This effect did not depend on the value of $\rho$ (*Figure 2D*).

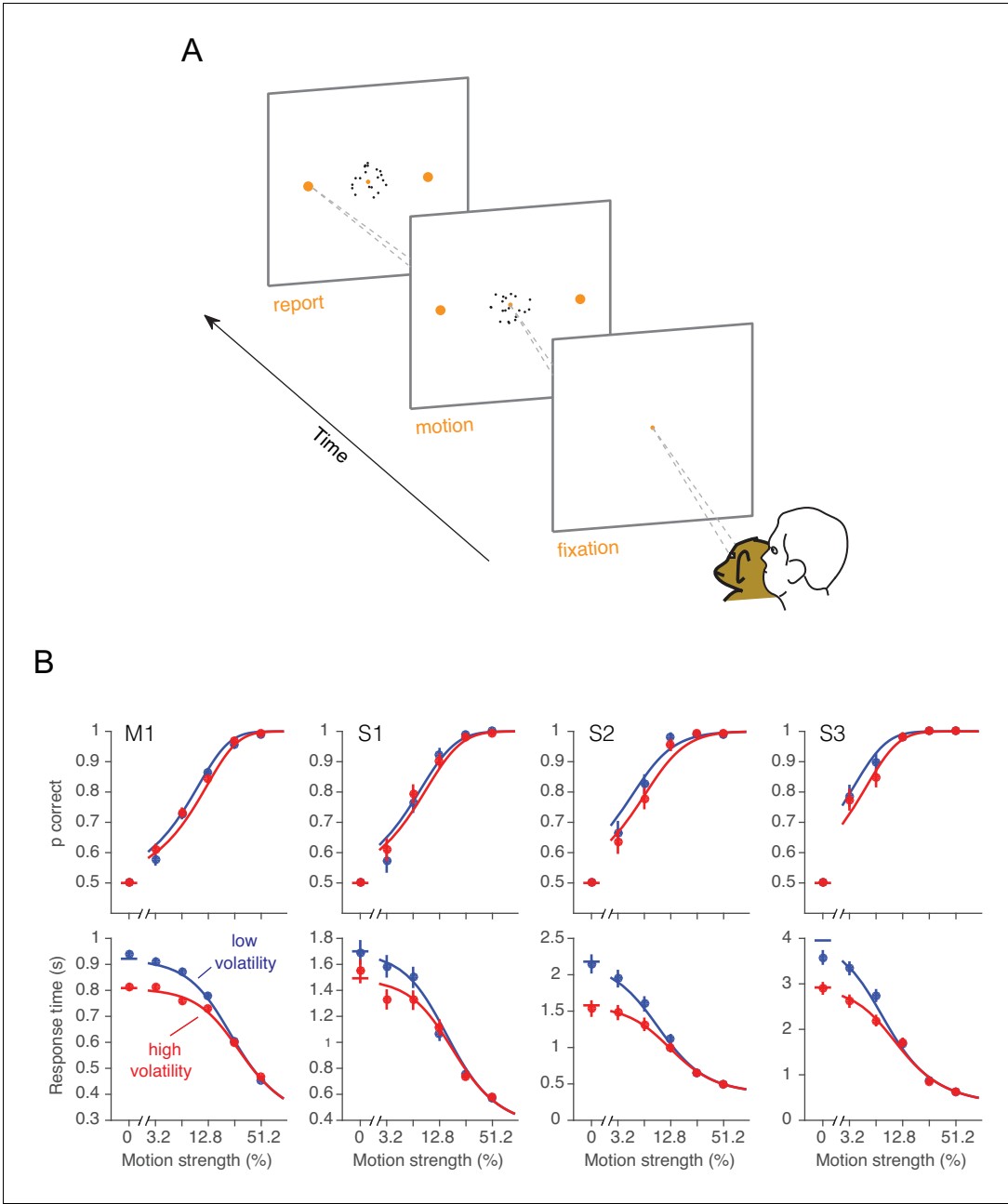

**Figure 3.** Effect of volatility on accuracy and reaction time. (**A**) Choice-reaction time task. One monkey and three humans were required to make a decision about the net direction of motion in a dynamic random dot display. Subjects reported their decision by making a saccadic eye movement to the right (left) target for rightward (leftward) motion. They could report their decisions at any time after the onset of motion. Trials of different coherences and volatilities were randomly interleaved. (**B**) Decision speed and accuracy. Each column represents a different subject. High volatility had only weak effects on accuracy (upper) but shortened the reaction times for all subjects (lower), particularly at the low motion strengths. Symbols are mean ± s.e. Solid traces are fits of a bounded evidence accumulation (drift diffusion) model. (M1, monkey; S1-S3, human subjects).

The following source data and figure supplement are available for figure 3:

**Source data 1.** Accuracy and reaction times from the choice-reaction time task.

**Figure supplement 1.** Accuracy in low and high volatility.

From these complementary analyses of stimulus and neural response, we conclude that the volatility manipulation has negligible effects on the expectation of momentary evidence and more substantial effects on the variance, especially at weak motion strengths. This enables us to proceed with a critical test of the bounded accumulation framework. In what follows we attempt to ascertain whether a change in the variance of the momentary evidence, introduced by our volatility manipulation, affects decision speed, accuracy, and confidence in accordance with the predictions of bounded evidence accumulation.

### Effect of volatility on choice and reaction time

One monkey (monkey W) and three humans were required to decide between two possible directions of motion and, when ready, to indicate their decision by looking to one of two targets (*Figure 3A*). For both high and low volatility conditions, stronger motion led to faster and more accurate choices. The main effect of high volatility was to decrease RTs, particularly at the weakest motion strengths (*Figure 3B*, bottom row, red). This effect was robust for all three human subjects and the monkey (*Equation 16*, all $p<0.03$, t-test, H$_0$: $\beta_2 = 0$). The manipulation affected the accuracy only subtly, and this was not statistically reliable for individual subjects in the RT task (*Figure 3B*, top row; for the four subjects: p=[0.35, 0.65, 0.2, 0.26], *Equation 17*, likelihood-ratio test). However, there was a significant effect of volatility on accuracy when pooling data across subjects and including data from the confidence tasks described below (*Equation 18*, $p<0.0005$, likelihood ratio test, H$_0$: $\beta_2 = 0$; see also *Figure 3—figure supplement 1*).

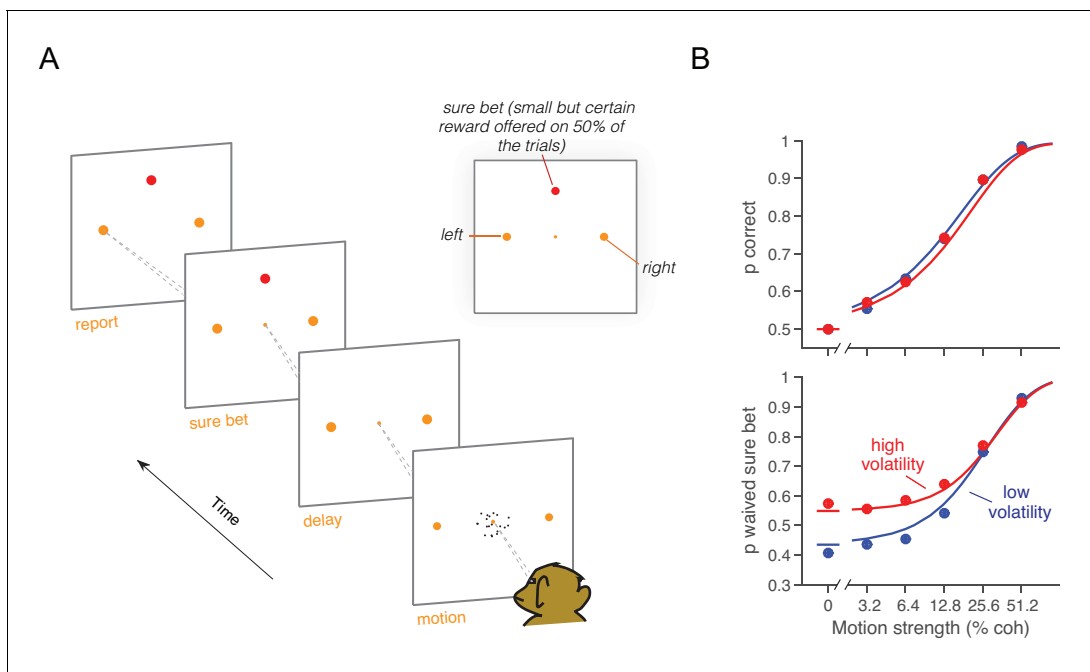

**Figure 4.** Effect of volatility on post-decision wagering. (**A**) Task. The monkey viewed the motion display for random duration, controlled by the computer, followed by a memory-delay period. When the fixation point was extinguished, the monkey indicated the net direction of motion by making an eye movement to a left or right choice target in order to receive a liquid reward, if correct. On a random half of trials, the monkey was presented a third 'sure bet' option (red target) during the delay period, which if chosen resulted in a small but certain reward. (**B**) Decision confidence and accuracy. Volatility did not affect accuracy systematically (upper), but the monkey waived the sure bet option more often on trials employing the high volatility display (lower), indicating greater confidence. The effect was concentrated at weak and intermediate motion strengths. Standard errors are shown but are smaller than the symbols. Solid traces are model fits (see Materials and methods).

The following source data and figure supplement are available for figure 4:

**Source data 1.** Proportion of correct and waived direction choices as a function of motion strength and volatility condition in the PDW task.

**Figure supplement 1.** Accuracy and PDW behavior as a function of stimulus duration and sure-target availability.

The pattern of results in *Figure 3B* is consistent with the hypothesis that decisions are made when an accumulation of noisy evidence reaches a bound. Indeed, the smooth curves are fits of this model to the data, where the variance of the momentary evidence is the only parameter that we allowed to change between conditions of high and low volatility (see below).

The effect of increased volatility on RT is most apparent at motion strengths near zero, for two reasons: (*i*) the volatility manipulation has a larger impact on variance of the motion energy at the weak motion strengths (*Figure 2B*), and (*ii*) the time to reach a bound is dominated by the variance of the momentary evidence, $\sigma_\Delta^2$ , when the motion strength is weak. For instance, when c = 0, the average time required by a diffusion process to reach a bound is proportional to $\sigma_\Delta^{-2}$ (*Shadlen et al., 2006*). These considerations also help to reconcile the contrast between the striking effects of volatility on RT versus subtle effects on choice accuracy: the volatility manipulation mainly affects the weakest motion strengths where accuracy is already poor (but see *Figure 3—figure supplement 1*). The important point is that by increasing noise, the volatility manipulation accelerates the dispersion of the decision variable away from its expected value and nearer the termination bounds, hence faster RT. A similar idea guides intuitions about the effect of volatility on confidence in a decision.

## Effects of volatility on confidence

Confidence refers to the belief that a decision one is about to make (or has just made) is likely to be correct. In the framework of bounded evidence accumulation, it can be formalized as the conditional probability of a correct choice given the state of the DV, which comprises the accumulated evidence and elapsed decision time (*Equation 5*). For the motion discrimination task, this can be calculated by considering, for each possible state of the DV, the likelihood that it was the result of motion strength of the appropriate sign. We refer to this as a mapping between DV and probability correct (*Figure 1C*). It depends on the set of possible motion strengths (the prior distribution of *c*), the two

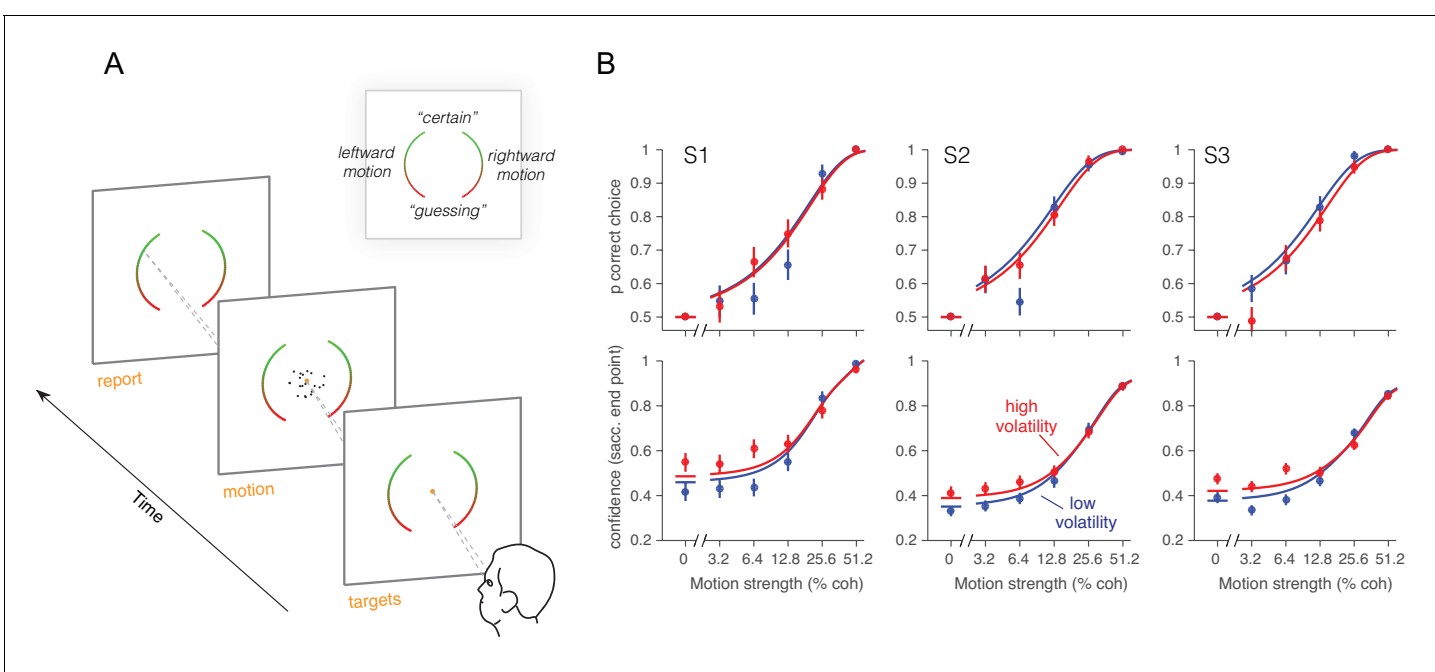

**Figure 5.** Effect of volatility on confidence rating. (A) Task. Subjects viewed random dot motion for 200 ms and subsequently indicated a direction decision and confidence rating by looking at a left or right target (circular arc). The position along the arc indicated confidence (inset). (B) Decision confidence and accuracy. Volatility again did not affect accuracy systematically (upper panels), but the three subjects issued higher confidence ratings on trials using the high volatility display. The effect was concentrated on the weak motion strengths. Symbols are mean ± s.e.; solid traces in the upper panels are model fits in which all but one parameter were fixed by the fits in *Figure 3B*. Solid traces in the lower panels are predictions.

The following source data is available for figure 5:

**Source data 1.** Decision confidence and accuracy in the human confidence task.

possible volatility conditions, and the amount of time that has elapsed in the trial. We assume the subject has implicit knowledge of this mapping, and does not adjust it when a low or high volatility stimulus is shown. The latter seems justified because volatility levels were randomly interleaved and not cued or even mentioned to the subjects (we evaluate this assumption, below, in several alternative models).

Increased volatility should affect confidence because it mimics an increase in the diffusion rate. At low coherences in particular, its main effect on the DV is to accelerate its exodus away from neutral (probability correct = 0.5) to more extreme values. Therefore, we predicted that volatility would increase confidence at low coherences, for the same reason that it speeds the RT. To test this prediction, we used two variants of the motion task, tailored to the abilities of monkeys and humans.

Monkey D was trained on a motion discrimination task with post-decision wagering (*Kiani and Shadlen, 2009*) (PDW; *Figure 4A*). The monkey had to decide between two opposite directions of motion and report its decision after a memory delay. The monkey was rewarded for correct decisions and randomly on the 0% coherence trials. On half of the trials, the monkey had the opportunity to opt out of reporting the direction choice and to select instead a smaller but certain reward. The 'sure bet' option was not revealed until at least one-half second after motion offset (i.e., during the delay). The task design thus encouraged the monkey to perform the direction discrimination on every trial. After extensive experience with the standard RDM (>100,000 trials; low volatility condition), we introduced the high volatility RDM on a random half of the trials. Single- and multi-unit recordings during performance of this task furnished the data for *Figure 2C–D*, as well as additional neurophysiological analyses described later.

In both low and high volatility conditions, the monkey made rational use of the sure bet, opting out more often for weaker motion (*Equation 19*, $p<10^{-6}$, logistic regression, likelihood-ratio test; *Figure 4B*) and for briefer stimuli (*Equation 19*, $p<10^{-6}$, logistic regression; *Figure 4—figure supplement 1*). When the sure bet was offered but waived, choice accuracy was higher than when the sure bet was not offered (*Equation 20*, $p<10^{-6}$, logistic regression; *Figure 4—figure supplement 1*). This indicates that the monkey was more likely to opt out of rendering its decision when the answer was more likely to be wrong. It implies that the decision to accept or waive the sure bet is based on the state of the evidence on the trial and not a general propensity associated with each motion strength (*Kiani and Shadlen, 2009*).

The main question we wished to address is whether the high volatility condition would elicit fewer sure-bet choices, consistent with greater confidence. As shown in *Figure 4B* (lower panel), the proportion of trials the monkey decided to waive the sure-bet option (deciding instead for a riskier direction choice) was greater on the high volatility trials (*Equation 19*, $p<10^{-6}$, likelihood-ratio test). Thus, high volatility increased the monkey's confidence, and did so despite a negligible effect on accuracy (*Figure 4B*, upper). Further, like its effect on RT, volatility affected PDW mainly when the motion was weak (*Figure 4B*, lower).

We confirmed the relationship between volatility and confidence in human participants. Instead of using PDW, we asked subjects to report their confidence on a scale from "feels like I'm guessing" to "certain I'm correct." The same three observers that performed the reaction time task participated in this second experiment. The RDM (low or high volatility, randomly interleaved) was displayed for a fixed 200 ms on each trial, after which they reported the perceived direction of motion (left or right) and the confidence in their decision. Participants reported the choice and the confidence rating by looking at a particular position on one of two elongated targets (*Figure 5A*), where the left or right target specified the motion choice and the vertical position was used to indicate confidence. They were allowed to adjust their gaze to the desired level before finalizing their combined choice and confidence report (*Figure 5A*). We thus encouraged subjects to use all available information in the 200 ms stimulus for both reports (*Van den Berg et al., 2016*). The results from the human observers were similar to those from the monkey. Naturally, subjects were more confident for high coherence stimuli (*Equation 21*, $p<10^{-6}$, t-test; *Figure 5*). They also reported higher confidence for the high volatility stimuli, and the effect was most apparent for the low coherence stimuli (*Equation 21*, $p<0.0004$, t-test).

**Table 1.** Parameter fits for the three tasks.

| Task | | RT Task | | | | PDW | Confidence task | | |
|---|---|---|---|---|---|---|---|---|---|
| Subject | | M1 | S1 | S2 | S3 | M2 | S1 | S2 | S3 |
| $\kappa$ | drift rate | 10.27 | 8.64 | 12.24 | 12.69 | 10.36 | 11.84 | 18.99 | 19.06 |
| $B_0$ | bound parameter | 1.96 | 1.26 | 1.47 | 1.97 | 2.23 | NA | NA | NA |
| a | bound parameter | 0.64 | −2.17 | −2.63 | −2.97 | NA | NA | NA | NA |
| d | bound parameter | −0.02 | −0.26 | −0.05 | −0.23 | NA | NA | NA | NA |
| $\mu_{tnd}$ | mean non-dec. time (s) | 0.28 | 0.35 | 0.34 | 0.38 | NA | NA | NA | NA |
| $\sigma_{tnd}$ | stdev non-dec. time (s) | 0.06 | 0.04 | 0.02 | 0.001 | NA | NA | NA | NA |
| $\phi$ | conf. separatrix | NA | NA | NA | NA | 0.63 | NA | NA | NA |
| $\beta$ | noise scaling param. | 1.10 | 0.69 | 2.21 | 2.19 | 1.55 | RT | RT | RT |
| $\alpha$ | noise scaling param. | 0.34 | 0.10 | 0.33 | 0.47 | 0.56 | RT | RT | RT |
| $\gamma$ | noise scaling param. | 0.40 | 2.31 | 2.98 | 2.29 | 0.57 | RT | RT | RT |

NA: not applicable; RT: values extracted from the fits to the RT task.

## A common mechanism for the effects of volatility on choice, RT and confidence

So far, the effect of volatility has been described qualitatively. Now we show how a single bounded accumulation model can account for the combined effect of motion strength and volatility on choice, accuracy and RT. In the model, choice, RT and confidence result from the accumulation of noisy momentary evidence as a function of time, until the integral of the evidence (the decision variable, DV) reaches one of two bounds, or for the PDW and confidence tasks, until the stimulus is curtailed. In the latter case, the sign of the DV determines the choice.

The DV is updated at each time step by the addition of a constant, proportional to motion strength, plus a draw from a zero-mean Gaussian distribution. In the language of drift-diffusion, the former gives rise to deterministic drift and the latter to a Wiener process scaled by a diffusion coefficient. The noise is itself comprised of stochastic contributions from the stimulus and its neural representation. Many studies make the simplifying assumption that the variance of the momentary evidence is fixed and independent of motion strength (*Ditterich et al., 2003*; *Palmer et al., 2005*; *Shadlen et al., 2006*). This would be the case if the momentary evidence obeyed the idealization in *Figure 2B* and if the neural responses of rightward and leftward preferring neurons exhibited variance that scaled linearly with mean. Then the difference between population responses would have the same variance for all motion strengths. However, the partial rectification (*Figure 2C*) implies that the variance of the difference should increase as a function of motion strength.

We characterize the dependency of the diffusion coefficient on motion strength and volatility based on the empirical observations of *Figure 2*. These analyses showed that (i) the variance of the momentary evidence increases with motion strength, and (ii) the difference in noise between volatility conditions is larger at 0% coherence and decays gradually for stronger motion. We capture these observations with a simple parameterization of the diffusion coefficient (*Equations 2 and 3*). First, we assumed that in the low volatility condition, the variance of the momentary evidence increases linearly with motion strength (*Figure 2—figure supplement 1*, blue trace; note the log scale of the abscissa). Second, we modeled the additional variability introduced by the doubly-stochastic manipulation as a variance offset at 0% coherence that decays exponentially as motion strength increases (*Figure 2—figure supplement 1*, red trace).

The framework can explain confidence if we assume that the brain has implicit knowledge of (*i*) the state of the accumulated evidence, (*ii*) the elapsed deliberation time, and (*iii*) the mapping of time and evidence to the probability of making a correct choice (*Figure 1C*). Time matters because the same level of accumulated evidence is associated with lower levels of accuracy if the evidence was accrued over longer periods of time (*Figure 1C*). In PDW, a sure-bet choice supersedes a direction decision if the probability correct (estimated from the state of accumulated evidence and the

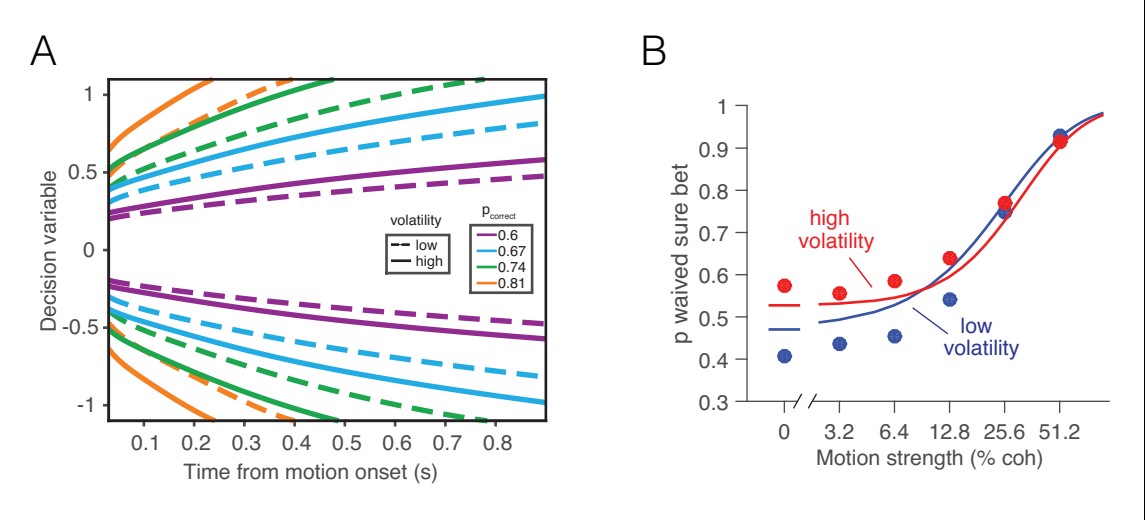

**Figure 6.** Separate mappings between the DV and confidence for high and low volatility do not explain post-decision wagering. High and low volatility conditions would confer a different correspondence between accumulated evidence and probability correct. (A) Iso-probability contours for the probability of a correct choice under low (dashed) and high (solid) volatility. For the same stimulus duration, a larger excursion of the decision variable is required under high volatility to reach the same level of expected accuracy. (B) Probability of waiving the sure bet as a function of motion coherence, shown separately for conditions of low and high volatility. Data points are the same as in *Figure 4*. Solid lines represent the best fitting 'two-map' model, which produce visibly worse fits than the model which relies on a single, common mapping for both volatility conditions (*Figure 4*). DOI: 10.7554/eLife.17688.017

decision time) is lower than a criterion Φ. In the human confidence task, probability correct is transformed into a confidence rating through a monotonic transformation (Materials and methods).

**Table 2.** Parameter fits for the alternative models.

| Task | | RT Task | | | | PDW | | |
|------|------|------|------|------|------|------|------|------|
| **Model description** | | **Different bound heights ($B_0$) for high and low volatility** | | | | **Two maps** | **Two maps, gradually** | **Two maps and two bounds** |
| *Subject* | | M1 | S1 | S2 | S3 | M2 | M2 | M2 |
| κ | drift rate | 10.56 | 8.71 | 10.76 | 12.31 | 10.72 | 10.40 | 10.44 |
| $B_0$ | bound parameter | 1.77 | 1.27 | 1.47 | 1.94 | 2.24 | 2.27 | 2.92 |
| $\Delta B_0$ | bound increase, high volatility | 0.17 | −0.06 | −0.14 | 0.17 | NA | NA | -1.0495 |
| a | bound parameter | 0.72 | −1.98 | −1.97 | −2.16 | NA | NA | NA |
| d | bound parameter | 0.31 | −0.33 | −0.07 | −0.47 | NA | NA | NA |
| $\mu_{tnd}$ | mean non-dec. time (s) | 0.28 | 0.35 | 0.33 | 0.37 | NA | NA | NA |
| $\sigma_{tnd}$ | stdev non-dec. time (s) | 0.056 | 0.037 | 0.02 | 0.001 | NA | NA | NA |
| φ | conf. separatrix | NA | NA | NA | NA | 0.626 | 0.628 | 0.629 |
| β | noise scaling param. | 1.04 | 1.11 | 0.82 | 2.37 | 1.99 | 1.60 | 1.96 |
| α | noise scaling param. | 0.673 | 0.003 | .0007 | 0.716 | 0.672 | 0.619 | 0.35 |
| γ | noise scaling param. | 0.66 | 1.76 | 1.79 | 0.94 | 0.32 | 0.415 | 0.16 |
| τ | Speed of volatility information accrual (s) | NA | NA | NA | NA | NA | 79.36 | NA |
| ΔBIC | Relative to base models | 29.42 | 7.7 | 27.3 | 12.1 | 252.4 | 7.24 | 126.9 |

NA: not applicable.

The solid curves in *Figures 3–5* are model fits. The model was fit to maximize the likelihood of the observables (choice and RT in the reaction time task; choice and sure bet in PDW). Best-fitting parameters are shown in *Table 1*. In the confidence task, we fit one parameter per subject ($\kappa$; see Materials and methods). This parameter was fit to maximize the likelihood of the direction choices. All other parameters were taken from the RT task, performed by the same participants. Therefore, the confidence curves in *Figure 5B* can be considered predictions of the model. These predictions capture the trend well, supporting the notion that time and accumulated evidence are the main determinants of confidence in a perceptual choice, even when noise is under experimental control. The overall quality of the fits—across all tasks and both species—indicates that the influence of motion strength and volatility on choice, reaction time and confidence can be explained by a common mechanism of bounded evidence accumulation.

## Alternative models

Up to now, we have attempted to explain the data on the assumption that subjects apply the same mapping between the accumulated evidence (the DV) and the probability that a decision rendered upon that evidence will be correct (i.e., confidence), regardless of the volatility condition. As stated earlier, the mapping is derived from all possible motion strengths, directions, and volatility conditions. Thus, we assume that subjects do not infer the noisiness of incoming evidence, or that if they do, they do not revise the mapping accordingly. An alternative is that the brain infers an estimate of the noisiness of the stimulus, in real time, to adjust the parameters of the decision process (*Deneve, 2012*; *Qamar et al., 2013*) or the evaluation of confidence (*Yeung and Summerfield, 2012*). This is a reasonable proposition, at least in principle, because the sample mean and variance of the motion energy can be used to classify volatility conditions with 90% accuracy (see Materials and methods).

We evaluated several 'two map' models which apply a different mapping between the DV and probability correct for each volatility condition. The first two-map model implements the assumption that subjects have full and immediate knowledge of the volatility condition on each trial. Although the maps are qualitatively similar (compare the iso-confidence contours of *Figure 6A*), the consequence of having separate maps is to reduce the effect of volatility on confidence. When fit to data, this two-map model produces visibly worse fits than the model that relies on a common map, despite having the same number of parameters (*Figure 6B*; $\Delta$BIC = 252.4 favoring the common-map model; see *Table 2* for parameter fits).

For the second two-map model, the assessment of volatility is not instantaneous but evolves over the course of a trial. For simplicity, we assumed that the probability of correctly identifying the volatility condition increases monotonically at a rate determined by a free parameter (see Materials and methods). Interestingly, the rate estimated from the best fit is exceedingly slow. For example, after 1 s of viewing, the weight assigned to the appropriate volatility map is just 1%. In other words, the confidence is dominated by the common mapping, consistent with our assumption. The fit is indistinguishable from the common-map model depicted in *Figure 4* (see *Table 2*), and the BIC statistic revealed that the addition of the extra parameter was not justified ($\Delta$BIC = 7.24).

We also considered the possibility that subjects used different termination criteria (bound heights) on low and high volatility trials. For the PDW task, this amounts to the addition of an extra free parameter in the first two-map model above. This model was also inferior to the simpler common-map model ($\Delta$BIC = 127; see *Table 2* for parameter fits). This is not surprising because in the PDW task, stimulus duration is controlled by the experimenter, and bounds merely curtail the expected improvement in accuracy on longer duration stimuli. We also fit a model for the RT task that allowed the bounds to be different for the two volatility conditions. This led to a marginal increase in the likelihoods, but not enough to justify the addition of the extra parameter ($\Delta$BIC = [29.4, 7.7, 27.3, 12.1] for the four subjects; *Table 2*).

These analyses of alternative models support our assumption that subjects applied a common mapping and decision strategy on trials of low and high volatility. We do not believe this holds generally but is likely a consequence of the particular volatility manipulation and task designs we employed. Indeed, the normative strategy for several model tasks, which approximate those in our study, would apply different bounds and mappings to the two volatility conditions (see Appendix). The full normative solution for the tasks we used is not known. Hence, we do not know if our

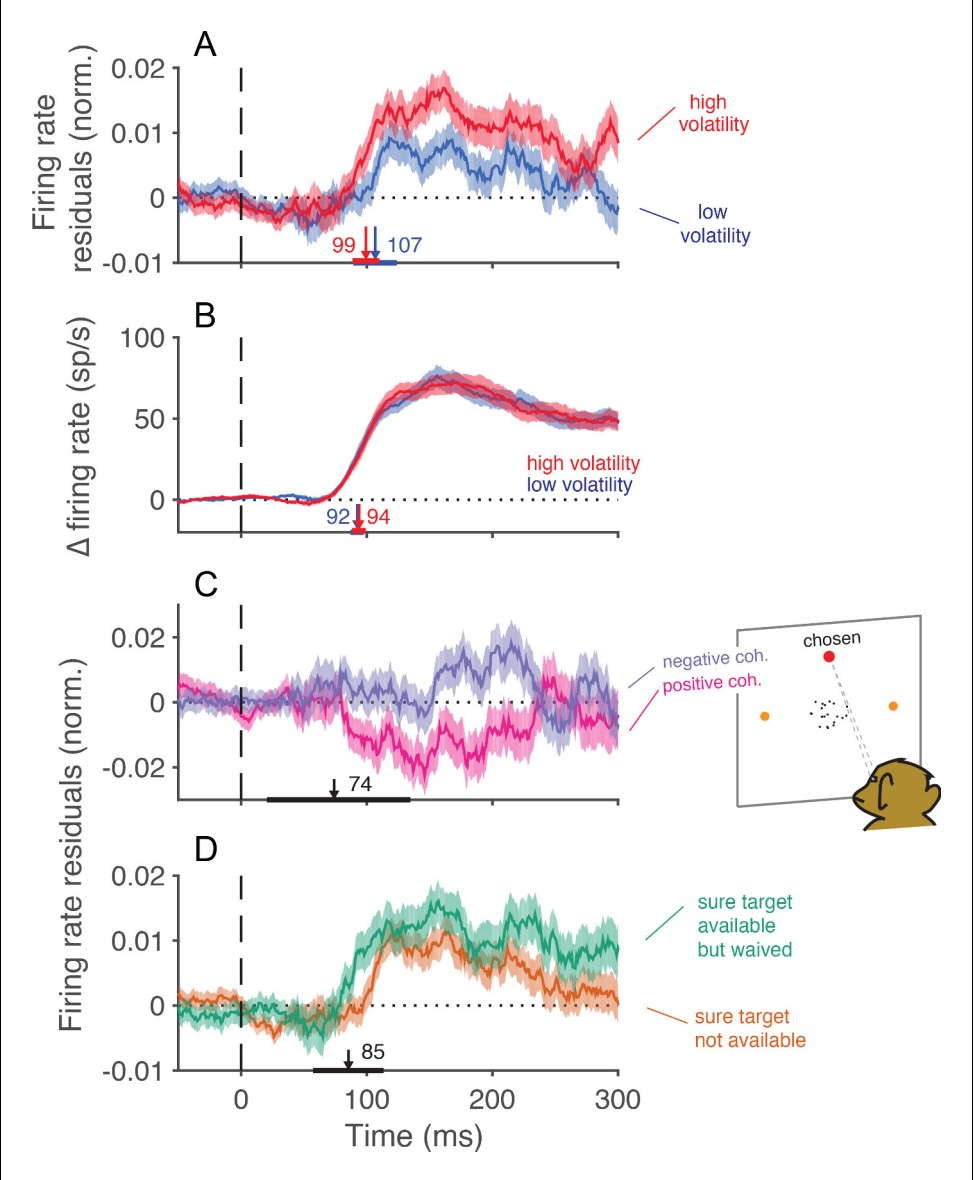

**Figure 7.** Trial-to-trial covariation between neural activity and behavior. (**A**) Average of the firing rate residuals, sorted by choice. The residuals are obtained by subtracting the average response to each motion strength and direction from the smoothed single-trial response. Positive values denote higher than average activity in support of the chosen alternative. Separate averages are shown for low (blue) and high (red) volatility trials. The vertical arrows show the time when the curves first differ from baseline, estimated with a curve-fitting procedure (see Materials and methods). The associated horizontal lines are ± one standard error of the latency estimates (bootstrap). Shading shows ± 1 s.e.m. across trials. (**B**) Difference in firing rate between the response to the preferred and the anti-preferred direction, for high-coherence trials. Trials of low and high volatility are shown in blue and red, respectively. Error bars represent s.e.m. across neurons. (**C**) Average of the firing rate residuals for trials in which the sure-bet target was chosen. For statistical power, we grouped trials from both volatility conditions. Neural responses are lower than average when the correct target is in the neurons' preferred direction (positive coherences, magenta), and above average when the motion is in the non-preferred direction (negative coherences, indigo). The arrow indicates the time at which the average residuals become significantly different from each other. (**D**) Average firing rate residuals sorted by choice, shown separately for trials where the sure bet was (green) or was not (orange) available. The average is greater when the sure bet was available but waived, consistent with the notion that the monkey waives the sure-bet target more often when the evidence appears to be stronger. The latency estimate (arrow) indicates the time that the difference between the curves becomes significant, which is similar to the time at which the neural activity is informative of the monkey's choice (**A**).

subjects performed suboptimally or if they were simply unable to identify the volatility conditions without adding additional costs (e.g., effort and/or time).

## Choice- and confidence-predictive fluctuations in MT/MST activity

The role of the neural data in this study was to validate and characterize the volatility manipulation in a population of neurons known to represent the momentary evidence used to inform decisions and confidence (*Salzman et al., 1990*; *Ditterich et al., 2003*; *Hanks et al., 2006*; *Fetsch et al., 2014*). Nevertheless, there are features of this limited data set which are germane to findings associated with the confidence task in particular. We share them in *Figure 7*, accompanied by the proviso that the data set is limited.

Consistent with earlier reports (*Britten et al., 1992*), trial to trial variation in the activity of neurons in MT/MST were indicative of the choice that the monkey was about to make. *Figure 7A* shows averaged residual responses, formed by subtracting the mean response for each motion strength as a function of time and multiplying by ±1 if the monkey chose the preferred of anti-preferred direction, respectively. Positive residuals therefore indicate an excess of activity in the chosen direction. For both low and high volatility conditions, trial-to-trial variation in the neural response was reflected in the monkey's choices. The fluctuations were more informative in the high volatility condition, presumably because they were induced by exaggerated variance in the motion display itself (e.g., *Figure 2A*). Notably, the time course of choice-related signals evolved with similar latencies in the low and high volatility conditions. The latencies were comparable to that of the direction selective signal itself (*Figure 7B*), suggesting that the choice was informed by the earliest motion information available in the stimulus (*Kiani et al., 2008*). The influence of neural variation declines over 200 ms, consistent with the idea that the brain terminates some decisions before the end of the stimulus presentation (*Kiani et al., 2008*).

The trial-by-trial variation in neural activity was also correlated with the decision to accept or waive the sure-bet option, when it was offered. Monkeys should opt out of the direction decision when the evidence is weak, and waive the sure bet when the evidence is strong. For positive coherences (i.e., net motion in the preferred direction), the residuals of firing rate were on average negative (*Figure 7C*, magenta trace). This implies that the monkey tended to opt out of the direction decision when the neural representation of the evidence was weaker than average. For negative coherences (net motion in the non-preferred direction), the residuals were positive on average (*Figure 7C*, blue trace), for an analogous reason. The difference between the two traces furnishes an estimate of the time course over which MT/MST neurons inform the decision to opt out. Notice the similarity in the time course of the choice and confidence signals (compare *Figure 7A and C*). The latency estimate derived from *Figure 7C* was unreliable (arrow and horizontal error bar, *Figure 7C*), but it was corroborated by a complementary analysis of the trials in which the monkey waived the sure bet (*Figure 7D*). Here we compared the average firing rate residuals on trials when the monkey waived the sure-bet option (green trace) with those on trials when the sure bet was not available (orange trace). We expect these traces to differ if the monkey waves the sure bet on trials when the neural responses are stronger. The point of divergence of the two traces in *Figure 7D* furnishes a more reliable estimate of the latency with which confidence signals are represented in the neuronal response (arrow). These results indicate that early motion evidence simultaneously informs both choice and confidence (*Zylberberg et al., 2012*). They are inconsistent with the proposal that choice and confidence are resolved in strict succession, as these predict that confidence selectivity ought to emerge later than choice-related signals (*Pleskac and Busemeyer, 2010*; *Navajas et al., 2016*).

## Discussion

We have shown that a stimulus manipulation that increases the variance of the momentary evidence bearing on a decision—what we term volatility—increases both the speed of the decision and the confidence associated with it. Testing the influence of volatility on the decision process is difficult, because it requires independent control over the signal and the noise in the evidence. We mimicked a manipulation of noise by changing the statistical properties of a dynamic stimulus. Our approach differs from recent studies that have attempted to vary evidence reliability through stimulus manipulations (*de Gardelle and Summerfield, 2011*; *Zylberberg et al., 2014*; *de Gardelle and Mamassian, 2015*) in that we (*i*) applied the manipulation to a well studied motion task for which much is

known about the underlying physiology; (*ii*) verified the effect of the manipulation by recording from neurons in the visual cortex of the macaque, and (*iii*) showed how a framework based on the bounded accumulation of evidence can account for the joint effect of volatility on choice, reaction time and confidence.

The modeling framework pursued here was able to explain the observed pattern of choices, RTs and confidence in a quantitatively coherent way (*Figures 3–5*), even predicting subjects' confidence ratings (*Figure 5B*) based on a fit to their RT data from a separate experiment (*Figure 3B*). The intuition is that increased volatility disperses the decision variable away from its expectation. For low coherences, it accelerates departure from the starting point (i.e., neutral evidence) and closer to one of the decision bounds. This tendency to arrive at larger absolute values of accumulated evidence— in support of either choice—leads to faster and more confident decisions (*Zylberberg et al., 2012*; *Maniscalco et al., 2016*). The intuition would apply to any theoretical framework that would associate confidence with the absolute deviation of a DV from neutral. This includes models based on signal detection theory (*Clarke et al., 1959*; *Ferrell and McGoey, 1980*; *Macmillan and Creelman, 2004*; *Kepecs and Mainen, 2012*; *Fleming and Lau, 2014*); however, these models ignore the temporal domain and are thus unable to account for RT or the strong correlation between deliberation time and confidence (*Figure 4—figure supplement 1*) (*Henmon, 1911*; *Pierrel and Murray, 1963*; *Vickers et al., 1985*; *Link, 1992*; *Kiani et al., 2014*).

These intuitions and our fits to the data rest on the assumption that subjects do not change their decision strategy based on the volatility of the evidence on a particular trial. On all trials, we assumed subjects applied the same termination policy (i.e., decision bound) and the same mapping between the state of the evidence and confidence, for both volatility conditions as well as for all motion strengths (*Gorea and Sagi, 2000*; *Kiani and Shadlen, 2009*). We considered and rejected alternative models in which the brain uses volatility to adjust the mapping and/or the decision bound. In particular, if different mappings between DV and confidence were used for the low and high volatility conditions, a larger excursion of the DV would be required in the high volatility condition to reach the same level of confidence, predicting a pattern of post-decision wagering behavior that was not supported by our data (*Figure 6*). In the RT task, volatility could be used to adjust the height of the decision bound in the face of lower reliability in order to maximize reward rate (*Deneve, 2012*; *Drugowitsch et al., 2014*). Indeed, the normative solution for a simplified version of the RT task is to increase the bound height on high volatility trials, which nevertheless leads to slightly faster responses than for low volatility trials when the motion is weak (*Appendix 1—figure 1*). However, this idea presupposes knowledge of reliability on the trials, which ought to predict lower confidence in the high volatility condition. Thus, models that posit an online estimation of reliability [cf., *Deneve (2012)*; *Yeung and Summerfield (2012)*; *Qamar et al. (2013)*] make predictions that run counter to one or more of the trends we observed.

This does not mean humans and monkeys are incapable of using information about stimulus reliability or difficulty to adjust their decision policy, and perhaps they would have in other circumstances (*Qamar et al., 2013*; *Shen and Ma, 2016*). For instance, had we used only a very difficult and a very easy condition, there would be a stronger incentive to ascertain the difficulty of the decision online and use different termination criteria for each condition. However, our experiment—in particular, the mixture of interleaved motion strengths and the volatility manipulation—is representative of a broad class of decisions for which the reliability of the evidence is unknown to the decision-maker before beginning deliberation and not readily apparent from a small number of samples. In such circumstances, an estimate of reliability might be viewed as another decision, which would entail (i) specification of alternative hypotheses about reliability, (ii) defining which stimulus features constitute evidence bearing on these hypotheses, (iii) accumulating the relevant evidence, and (iv) specifying a termination criterion for this decision. Such an evaluation must balance the benefits derived from the use of reliability to adjust the parameters of the decision process trial by trial, with the associated cost in time and effort.

Even if subjects were cued explicitly about reliability, it is not clear that they would adjust the decision criteria on a trial-by-trial basis. In a detection task where the stimulus categories were signaled by an external cue, human subjects did not adjust the decision criterion to the levels used when each stimulus category was presented on its own (*Gorea and Sagi, 2000*). Instead, subjects behaved as if they assumed a common distribution of signals encompassing all stimulus conditions and applied a single decision criterion. Our volatility manipulation was more subtle than an explicit

cue, but we do not doubt that our subjects could perform above chance in a 2AFC experiment if they were trained to identify the higher volatility stimulus among a pair sharing the same motion strength. If nothing else, they could monitor their own decision times and confidence. However, when a mixture of different levels of volatility are presented in a sequence of otherwise similar events (trials), subjects appear to combine trials of low and high volatility to form a single internal distribution with signed coherence as the only relevant dimension.

Our results highlight limitations to the brain's capacity to extract and exploit knowledge of volatility. Our study may therefore be of interest to psychologists and behavioral economists (*d' Acremont and Bossaerts, 2016*). Systems with multiple interacting units, like financial markets, sometimes give rise to 'leptokurtic' distributions, referred to as those where the probability of extreme events is larger than expected from normal distributions (*Mandelbrot, 1997*). A simple way of constructing leptokurtic distributions is by mixing Gaussian distributions that have the same mean but different variances, similar to our doubly stochastic (high volatility) stimulus. When interpreting 'leptokurtic' noise, people appear to overreact to outliers. For instance, when making stock investment decisions, people often misinterpret large fluctuations as evidence for a fundamental change in expected value (*De Bondt and Thaler, 1990*). Similarly, our subjects interpreted the 'outliers' introduced by our doubly stochastic procedure (motion bursts of unlikely strength given the average motion strength of the trial) as if they were caused by a higher coherence stimulus. In this sense, they behaved as if the noisy samples they acquired were generated by a mesokurtic distribution (e. g., Gaussian). Is intriguing to think that the inferences and biases that people display in simple decisions about stochastic motion may bear on how they interpret and act upon stochastic signals operating over longer time scales.

## Materials and methods

### Random dot stimuli

Three humans and two monkeys performed one or more tasks where they had to make binary choices about the direction of motion of a set of randomly moving dots drawn in a circular aperture. Dots could move in one of two opposite directions, and were generated as described in previous studies (e.g., [*Roitman and Shadlen, 2002*]). Briefly, three interleaved sets of dots were drawn in successive frames (monitor refresh rate: 75 Hz). When a dot disappeared, it was redrawn 40 ms later (i.e., 3 video frames) either at a random location in the stimulus aperture or displaced in the direction of motion.

We refer to trials where the probability of coherent motion is fixed within the trial as 'low volatility', and trials where it varies within the trial as 'high volatility'. Trials of low and high volatility were uncued and randomly interleaved. Example stimuli can be seen in *Video 1*.

### RT task

We studied the relationship between volatility and decision speed with a reaction-time version of the random-dot motion discrimination task (*Roitman and Shadlen, 2002*). Three human participants completed 6631 trials (subject S1: 2490 trials; S2: 2070; S3: 2071), and one macaque (monkey W) completed 14,137 trials.

Each trial started with subjects fixating on a central spot (0.33° diameter) for 0.5 s. Then two targets (1.3° diameter) appeared on the horizontal meridian at an eccentricity of 9° to indicate the two possible directions of motion. Observers had to maintain fixation for an additional 0.3–0.7 s (sampled from a truncated exponential with $\tau = 0.1$ s) and were then presented with the motion stimulus, centered at fixation and subtending 5° of visual angle. Dot density was 16.7 dots/deg$^2$/s, and the displacement of the coherent dots was consistent with apparent motion of 5 deg/sec.

Feedback was provided after each trial. Correct decisions were rewarded with a drop of juice (monkey) or a pleasant sounding chime (humans). Errors were followed by a timeout of 1 (human) or 5 (monkey) seconds, and, in humans, also accompanied by a low-frequency tone. For the monkey, a minimum time of 950 ms was imposed from dot onset to reward delivery (e.g., *Hanks et al., 2011*) in order to discourage fast guessing. Trials employing 0% coherence motion were deemed correct with probability ½.

## Confidence task (Monkey)

A second monkey (monkey D) was trained to perform a direction discrimination task with post-decision wagering (*Kiani and Shadlen, 2009*). After acquiring fixation, two targets appeared (6.5–9° eccentricity) to indicate the alternative directions of motion, followed by the motion stimulus after a variable time (truncated exponential; range 0.3–0.75 s, $\tau = 0.25$ s). Motion viewing duration was sampled from a truncated exponential distribution (range 0.1–0.93 s, $\tau = 0.3$ s). After motion offset, the monkey had to maintain fixation for another 1.2 to 1.7 s. During this delay, a third target (sure-bet target; $T_s$) appeared on half of the trials, no earlier than 0.5 s from motion offset, positioned perpendicular to the axis of motion. After this delay, the fixation point disappeared, cueing the monkey to report its choice. Correct decisions led to a juice reward, and incorrect decisions led to a timeout (5 s). Selecting the sure-bet led to a small but certain reward, roughly equivalent to 55% of the juice volume received in correct trials.

The monkey performed a total of 65,751 behavioral trials, a subset of which (44,334 trials) were accompanied by neurophysiological recordings. By convention, positive motion coherences correspond to the preferred direction of motion of the recorded neurons. When paired with neural recordings, the speed and direction of motion, and the size of the circular aperture, were adjusted to match the properties of the neuron or multiunit site under study (see below).

## Confidence task (Human)

The relationship between volatility and confidence was also studied in a task that required explicit confidence reports. After the subject fixated a central spot, two crescent-shaped targets appeared on each side of the fixation (*Figure 5*). The targets were the left and right arcs of a circle (radius 10° visual angle) centered on the fixation point. These arcs were visible for for $2\pi/3$ radians (i.e., extending $\pm 60°$ angle above and below the horizontal meridian). The left (right) target ought to be selected to indicate that the perceived direction of motion was to the left (right, respectively). Subjects were instructed to select the upper extreme of the targets if they were completely certain of their decision, and the lowermost extreme if they thought they were guessing. Intermediate values represent intermediate levels of confidence. Visual aid was provided by coloring the targets in green at the top, red at the bottom, with a gradual transition between the two. After a variable delay during which participants had to maintain fixation, the random dot motion stimulus was shown for a fixed duration of 200 ms. Dot speed, density and aperture size were identical to the RT experiment. After motion offset, the subjects were required to indicate their response by directing the gaze to one target. Decisions were reported without time pressure and subjects were allowed to make multiple eye movements until they pressed the spacebar to accept the confidence and the choice. The same participants that completed the RT task performed the confidence task (subject S1: 1536 trials; S2: 2103; S3: 2107).

## Neurophysiological methods

All animal procedures complied with guidelines from the National Institutes of Health and were approved by the Institutional Animal Care and Use Committee at Columbia University. A head post and recording chamber were implanted using aseptic surgical procedures. Multi- (MU) and single-unit (SU) recordings were made with tungsten electrodes (1–2 MΩ, FHC). Areas MT (n = 13 SU and 9 MU sites) and MST (n = 13 SU, 12 MU) were identified using structural MRI scans and standard physiological criteria. We did not observe substantial differences between the two areas in the main results (*Figure 2*) and therefore pooled the data for all analyses. However, the sample size is too small to rule out subtle differences between areas.

The electrode was advanced while the monkey viewed brief, high-coherence random-dot motion stimuli of different directions while fixating a central target. When we encountered an area with robust spiking activity and clear direction-selectivity, we attempted to isolate a single neuron (Sort-Client software, Plexon Inc., Dallas, TX, USA) but otherwise proceeded with mapping of receptive field position, size, preferred speed and direction based on multiunit activity, as described previously (*Fetsch et al., 2014*). When direction tuning was sufficiently strong (>2 S.D. separating firing rates for preferred vs. anti-preferred direction motion), we proceeded with the PDW task, tailoring the stimulus to the neurons' RF and tuning properties and aligning the choice targets with the axis of motion.

## Bounded accumulation model

Solid lines in *Figures 3–5* represent fits (or predictions) of a bounded accumulation model. In the model, noisy momentary evidence is accumulated until the integral of the evidence (termed the decision variable, DV) reaches one of two bounds at $\pm B(t)$, or until the motion stimulus is terminated by the experimenter. The momentary evidence comprises samples from a Gaussian distribution with mean $\kappa c$ and variance $\sigma_v^2(\mathrm{c})$, where $\kappa$ is a constant, $c$ is the motion coherence, and $v$ indicates whether the volatility is high or low. In most applications of diffusion models, the variance is assumed to be fixed and independent of motion strength, but our analyses of the motion energy and the neuronal recordings (*Figure 2*), motivate a more complex dependence of variance on $c$ and $v$. To capture these trends parsimoniously, we modeled the variance as a linear function of motion strength

$$\sigma_{low}^2(c) = 1 + \beta|c| \tag{2}$$

plus an offset for the high volatility, which was maximal at c = 0 and diminishing at higher coherences:

$$\sigma_{high}^2(c) = \sigma_{low}^2(c) + \alpha e^{-\gamma|c|} \tag{3}$$

The three degrees of freedom $(\beta, \alpha, \gamma)$ control the slope of the coherence dependence, the effect of volatility at $c = 0$, and its diminishing effect at higher coherence (*Figure 2—figure supplement 1*). We constrained the variance in the high volatility condition to be monotonically increasing. Note that the unity constant in *Equation 2* is necessary because a model in which the offset is a free parameter in addition to $\kappa$ and $B(t)$ is equivalent to one in which the offset is set to 1 and $\kappa$ and $B(t)$ are scaled appropriately (*Palmer et al., 2005*; *Shadlen et al., 2006*).

For a given motion coherence and volatility ($v$), the probability density function for the state of the decision variable ($x$) as a function of time ($t$) is given by a one-dimensional Fokker-Planck equation:

$$\frac{\partial p(x,t)}{\partial t} = -\kappa c \frac{\partial p(x,t)}{\partial x} + 0.5\sigma_v^2(c)\frac{\partial^2 p(x,t)}{\partial^2 x} \tag{4}$$

where $p$ is the probability density of decision variable $x$ at time $t$. Boundary conditions were such that the probability mass is 1 for $x = 0$ at $t = 0$, and the probability density vanishes at the upper and lower bounds $\pm B(t)$.

Confidence is given by the probability of being correct given the state of the evidence ($x$) and elapsed time, which could either correspond to the time of bound-crossing or the stimulus duration if no bound was reached. Because the direction decision depends on the sign of $x$, the sign of the decision variable must equal the sign of the coherence for the choice to be correct, except for 0% coherence trials that are rewarded at random. Therefore,

$$\begin{aligned} p(corr|x,t) &= \sum_v p(corr \mid x,t,v)p(v|x,t) \\ p(corr|x,t,v) &= \sum_{c|sign(c)=sign(x)} p(c|x,t,v) + \tfrac{1}{2}p(c=0|x,t,v) \end{aligned} \tag{5}$$

where $t$ is either the time at which the bound was hit or the time at which the stimulus was curtailed. The distribution over coherences $p(c|x,t,v)$ can be obtained by Bayes rule, such that $p(c|x,t,v) \propto p(x,t|c,v)p(c|v)$, where the constant of proportionality ensures that $\sum_c p(c|x,t,v) = 1$. This constitutes a mapping between the DV and probability correct, which is the basis for assignment of confidence to a decision (*Figure 1C*). In general we assume that the same mapping $p(corr|x,t)$ supports confidence ratings (and PDW) on all trials irrespective of volatility, but evaluate this assumption using the alternative models described below.

The data were fit to maximize the likelihood of the parameters given the choice, confidence and RTs observed on each trial. In the RT task, the model parameters were maximum likelihood fits to choice and RT:

$$\hat{\xi}^{RT} = \arg\max_{\xi^{RT}} \left( \sum_{i=1}^N log\big(p\big(choice_i, RT_i|c_i, v_i, \xi^{RT}\big)\big) \right) \tag{6}$$

where $\xi^{RT}$ represents the model parameters for the RT task, $i$ is the trial number and $N$ is the total number of trials. The probability density function for the time of bound crossing (decision times) is obtained by numerical solutions to the Fokker-Planck equation. The difference between the reaction time and the decision time is the non-decision latency, assumed to reflect sensory and motor delays unrelated to motion strength or volatility. This latency is assumed Gaussian with mean $\mu_{tnd}$ and standard deviation $\sigma_{tnd}$. The RT probability density function is obtained by convolving the p.d.f. of the decision times with the distribution of non-decision latencies.

For the PDW task, the log likelihood is a sum of two terms,

$$\hat{\xi}^{PDW} = \underset{\xi^{PDW}}{\arg\max}\left(L^{S^+} + L^{S^-}\right) \tag{7}$$

where $L^{S^+}(L^{S^-})$ is the log-likelihood computed over trials with (without) the sure-bet target, and $\xi^{PDW}$ are the model parameters. For trials without the sure target, the log-likelihood of the parameters is

$$L^{S^-} = \sum_{i=1}^{N} \log\left(p\left(choice_i | c_i, v_i, T_i, \xi^{PDW}\right)\right) \tag{8}$$

where the summation runs over trials without the sure target, and $T_i$ is the duration of the stimulus on trial $i$. The argument of the summations is computed as follows. If $p_{up}(t)$ is the probability of crossing the upper bound at time $t$, then the probability of crossing the bound anytime before time $T$ is

$$P_{up}\left(T | c, v, \xi^{PDW}\right) = \int_0^T dt\, p_{up}\left(t | c, v, \xi^{PDW}\right) \tag{9}$$

and

$$p\left(choice = 1 | c, v, T, \xi^{PDW}\right) = P_{up}\left(T | c, v, \xi^{PDW}\right) + p\left(x{>}0, t = T | c, v, \xi^{PDW}\right) \tag{10}$$

where choice '1' is associated with a positive DV (i.e., $x{>}0$). In the equation, $p\left(x{>}0, t = T | c, v, \xi^{PDW}\right)$ is the probability that the decision variable ($x$) is positive at time $T$ and that no bound has been reached before $T$.

For trials where the sure-bet target was offered, we compute the likelihood of the parameters given the three possible responses in a trial: the two directional choices and the sure bet choice. We assumed that subjects opt out of reporting the direction choice and select the sure bet if the confidence in the decision is lower than a criterion, $\Phi$, which was the same for conditions of low and high volatility. The value identifies a probability contour like those depicted in **Figure 1C**. It demarcates a zone in the middle of the graph depicted in **Figure 1C** in which the state of the evidence would lead the subject to opt out. Therefore, the probability of opting out of the direction choice $p(o)$ is

$$
\begin{aligned}
p(o | c, v, T, &\xi^{PDW}) \\
&= \int_{-B(t)}^{+B(t)} dx\, p(x, t = T | c, v, \xi^{PDW})\mathcal{H}(\Phi - p(corr | x, T)) \\
&+ \int_0^T dt\, p_{up}(t | c, v, \xi^{PDW})\mathcal{H}\left(\Phi - p(corr | B_{up}(t), t)\right) \\
&+ \int_0^T dt\, p_{lo}(t | c, v, \xi^{PDW})\mathcal{H}(\Phi - p(corr | B_{lo}(t), t))
\end{aligned} \tag{11}
$$

where $\mathcal{H}(x)$ is a step function that evaluates to one if $x{>}0$, and zero otherwise. The first term on the right-hand side of the equation integrates the probability density that has not been absorbed at a bound before time $T$ and for which probability correct is lower than $\Phi$. The second and third terms allow for the possibility that even when a bound was reached, the probability correct at the bound is lower than the criterion $\Phi$. In practice, this only occurs (e.g., during fitting) when the bound is too low or the criterion is too high. $B_{up}(t)$ and $B_{lo}(t)$ correspond to the height of the upper and lower bounds at time $t$, respectively. For readability, we have omitted the dependence of $p(corr)$ on some parameters (e.g., $\xi^{PDW}$).

The probability of waiving the sure bet and making a direction choice follows the complementary logic:

$$p(choice = 1|c,v,T,\xi^{PDW})$$
$$= \int_0^T dt\, p_{up}(t|c,v,\xi^{PDW})\mathcal{H}(p(corr|+B(t),t)-\Phi)$$
$$+ \int_0^{+B(t)} dx\, p(x,t=T|c,v,\xi^{PDW})\mathcal{H}(p(corr|x,T)-\Phi) \tag{12}$$

where the first term of the right-hand side corresponds to the probability of selecting choice '1' when the bound is reached, and the second term computes the probability of selecting this choice when no bound is reached before $T$.

In the human confidence task, we performed a maximum likelihood fit to the choice reported on each trial:

$$\hat{\xi}^{HCONF} = \arg\max_{\xi^{HCONF}}\left(\sum_{i=1}^N \log\big(p\big(choice_i|c_i,v_i,T_i,\xi^{HCONF}\big)\big)\right) \tag{13}$$

where $\hat{\xi}^{HCONF}$ is the maximum likelihood estimate of the parameters and the likelihood is computed as described by *Equation 10*. We fit only one parameter per subject ($\kappa$). The rest of the parameters were taken from the RT task (i.e., from $\hat{\xi}^{RT}$; see *Table 1*). Note that confidence was not used for the fits, and therefore the solid curves in *Figure 5* can be considered predictions of the model.

For the RT task, we allowed the bound height to change as a function of time, as suggested by previous work (*Churchland et al., 2008*; *Hanks et al., 2011*; *Drugowitsch et al., 2012*). The upper and lower bounds were symmetric around zero, and were parameterized by a logistic function of time:

$$B(t|a,d) = \pm B_0\left(1+\exp^{a(t-d)}\right)^{-1} \tag{14}$$

where $a$ and $d$ are the scale and location parameters of the logistic. The bound parameters were constrained to be the same for the two volatility conditions, except in the alternative model for the RT task where we fit separate $B_0$ for the two volatility conditions (*Table 2*).

In the human confidence task, the presence of bounds did not improve the quality of the fits. This implies that subjects used all the stimulus information to inform their choices, presumably because the stimulus duration was only 0.2 s. In the PDW, a stationary bound (i.e., $B(t) = B_0$) improved the quality of the fits.

In the human confidence experiment, we do not know how each subject maps a position on the rating scale (position along the crescent target) to probability correct. Therefore, we assumed a monotonic transformation between the expected probability correct $p(corr|c,v)$ and saccadic end point. Probability correct $p(corr|c,v)$ was obtained by marginalizing $p(corr|x,t)$ over the state of the evidence ($x$) at the time of decision termination ($t$). Because we did not include a bound in the human confidence task, $t$ is the stimulus duration (i.e., $T = 0.2$ s). The distribution of the DV at decision time depends on coherence $c$ and volatility $v$, therefore

$$p(corr|c,v) = \int dx\, p(corr|x,T)\, p(x,t=T|c,v) \tag{15}$$

The monotonic transformation $\mathcal{F}$ that maps probability correct to the average position in the rating scale $\langle sac(c,v)\rangle_{tr}$ was constructed as a linear combination of three error functions plus a constant offset: $\mathcal{F}(x) = \sum_{i=1}^3 w_i\, erf_i\left(\frac{x-o_i}{s}\right)+k$, where $o_i$ is an offset term, and $s$ is a scaling parameter. The three linear weights and the offset $k$ were fit to minimize the sum of squared differences between $\mathcal{F}[p(corr|c,v)]$ and $\langle sac(c,v)\rangle_{tr}$. Similar results were obtained using different parameterizations of $\mathcal{F}$.

For the PDW task, we explored three alternative 'two map' models. In the first, we used a different mapping between DV and confidence for each volatility condition. Each map is the one that should be used if the volatility condition of each trial were known (i.e., the one specified by the bottom row of *Equation 5*). For the second two-map model, the assessment of volatility develops

gradually during the trial. We assume that for a trial $i$ with stimulus duration $T_i$, the probability that the decision maker can identify the trial's volatility is given by $w(T_i) = 1 - e^{-T_i/\tau}$. For trials where the sure bet was offered, we compute the probability of the action that was chosen by the monkey as a weighted average of the two probabilities: the probability that results from using a common map for both volatility conditions, which was weighted by $(1 - w(T_i))$, and the probability obtained from using the mapping that corresponds to the appropriate volatility of the trial, which was weighted by $w(T_i)$. The time constant $\tau$ was fitted to data. If $\tau$ is small, information about volatility builds up rapidly and the decision maker can use the appropriate map for each condition. Fitting the model to data showed that the volatility information develops very gradually, with $w(t)$ being ~0.01 for a 1-s stimulus. For the third model, besides using different mappings between DV and confidence for the two volatility conditions, we also fit independent bounds, such that $B_0^{high} = B_0 + \Delta B_0$ where $B$ denotes bound height (see *Table 2*). Best fitting parameters for the three alternative models and the BIC comparisons to the model of *Figure 4* are shown in *Table 2*.

## Statistical analysis

To examine whether high volatility leads to faster responses in the reaction time task, we fit a linear regression model for each subject where the reaction time is given by

$$RT = \beta_0 + \beta_1|c| + \beta_2 I_v \tag{16}$$

where $I_v$ is an indicator variable for volatility (1: high, 0: low), and $\beta$'s are fitted coefficients. Unless otherwise indicated, the null hypothesis is that the $\beta$ term associated with $I_v$ equals zero, evaluated with $t$-test ($t$-statistics were derived using the parameter estimates and their associated standard errors [i.e., the square root of the elements in the diagonal of the covariance matrix of the parameter estimates]).

To evaluate the influence of volatility on accuracy, we used logistic regression, excluding trials of 0% coherence:

$$p_{correct} = \left[1 + e^{-(\beta_0 + \beta_1|c| + \beta_2 I_v)}\right]^{-1} \tag{17}$$

The influence of volatility was evaluated with a likelihood-ratio test comparing models with and without the $\beta_2$ term.

We also used logistic regression to evaluate the effect of volatility on accuracy when pooling data across subjects and experiments:

$$p_{correct} = \left[1 + e^{-\left(\beta_{0,s,x} I_{s,x} + \beta_1|c| + \beta_{2,s,x} I_{s,x} I_v\right)}\right]^{-1} \tag{18}$$

where $I_{s,x}$ are indicator variables for every combination of task and subject (n = 8). This equation parallels the structure of the previous one. The first term in the argument of the exponential allows fitting a different intercept for each combination of task and subject, and the third term allows for different intercepts on high and low volatility trials. The significance of the influence of volatility on accuracy was evaluated with a likelihood ratio test comparing nested models with and without the $\beta_2$ terms, with the test statistic evaluated against a $\chi^2$ distribution with n = 8 degrees of freedom. Only non-zero coherences were included in this analysis.

Similarly, to evaluate the influence of volatility on the monkey's PDW behavior on trials where the sure bet was offered, we fit

$$p_{waived} = \left[1 + e^{-(\beta_0 + \beta_1|c| + \beta_2 I_v + \beta_3 T_d)}\right]^{-1} \tag{19}$$

where $p_{waived}$ is the probability that the sure bet was declined, and $T_d$ is stimulus duration. We also examined whether availability of the sure bet influenced accuracy:

$$p_{correct} = \left[1 + e^{-(\beta_0 + \beta_1|c| + \beta_2 I_v + \beta_3 T_d + \beta_4 I_s|c|)}\right]^{-1} \tag{20}$$

where $I_s$ is 1 if the sure bet was offered, and 0 otherwise. A positive $\beta_4$ indicates that the accuracy increases if the sure bet is offered but waived.

In the human confidence task, we mapped subjects' confidence reports to a 0–1 scale, such that '0' stands for 'guessing' and '1' for 'full certainty'. To evaluate the significance of the effect of volatility on confidence we fit for each subject the following linear regression model:

$$conf = \beta_0 + \beta_1|c| + \beta_2 I_v \qquad (21)$$

## Motion energy

While the motion coherence specifies the nominal strength of motion in the stimulus, the effective motion strength varies from trial to trial and even within trials, due to the random fluctuations in the stimulus. To extract the effective motion strength, we computed the motion energy in the stimulus (**Adelson and Bergen, 1985**; **Kiani et al., 2008**), following published procedures which we briefly review here. We convolved the sequence of random dots presented on each trial with two pairs of spatiotemporal filters. Each pair of filters is selective for one of the two alternative directions of motion ($\pm x$). Directional selectivity is achieved through the addition or subtraction of two space-time separable filters. As in previous work (**Kiani et al., 2008**), the temporal impulse responses are:

$$
\begin{aligned}
fast(t) &= (60t)^3 \exp(-60t)\left[\tfrac{1}{3!} - \tfrac{(60t)^2}{(3+2)!}\right] \\[6pt]
slow(t) &= (60t)^5 \exp(-60t)\left[\tfrac{1}{5!} - \tfrac{(60t)^2}{(5+2)!}\right]
\end{aligned}
\qquad (22)
$$

The spatial filters are even (mirror-symmetric) and odd (non-symmetric) fourth order Cauchy functions:

$$
\begin{aligned}
even(x,y) &= \cos^4(\alpha)\cos(4\alpha)\exp\left(-\tfrac{y^2}{2\sigma_g^2}\right) \\[6pt]
odd(x,y) &= \cos^4(\alpha)\sin(4\alpha)\exp\left(-\tfrac{y^2}{2\sigma_g^2}\right)
\end{aligned}
\qquad (23)
$$

where $\alpha = tan^{-1}(x/\sigma_c)$. The constants in **Equations 22 and 23** were adjusted to match the apparent speed of the coherently moving dots.

The two pairs of directionally selective filters were obtained through appropriate addition and subtraction of the product of a spatial and a temporal filter. Specifically, the two filters selective to the +x direction are given by 'slow × even – fast × odd', and 'slow × odd + fast × even'. Filters selective to the -x direction are given by 'fast × odd + slow × even', and 'fast × even – slow × odd'. The four directional filters were convolved with the 3-dimensional (x,y,time) stimulus. After squaring the output and adding the two filters that prefer the same direction, we compute opponent motion energy by subtracting -x from +x preferring responses. Finally, we average across space to obtain a temporal signal, $e_{tr}(t)$, which quantifies how motion strength varies within each trial. Because the motion energy has arbitrary units, which varies, for instance, with the size of the stimulus, we normalized $e_{tr}(t)$ multiplying it by a constant $\lambda$. The normalization constant was the same for all trials in a session, and was set such that the motion energy is, on average, equal to the motion coherence. This normalization is possible because the motion energy is a linear function of the motion coherence. The motion energy profile for $e_{tr}(t)$ is shown in **Figure 2A** for an example trial.

To characterize the mean and variance of the motion energy for high and low volatility (**Figure 2B**), we first computed the average motion energy for each trial, i.e. $e_{tr} = \langle e_{tr}(t)\rangle_t$, ignoring the rise and decay times of the motion filters, that is from 50 ms after motion onset to 50 ms after offset. The mean and variance of $e_{tr}$ was computed over subsets of trials grouped by motion coherence and volatility condition.

We used logistic regression to determine if the motion energy profile of each trial of the PDW task contains enough information to identify the trial's volatility. We calculated the mean ($e_{tr}$) and an index of the dispersion ($e_{tr}^v$) of the motion energy time course for each trial. The dispersion index was estimated as the variance of the distribution of motion energy values estimated at the frame rate, ignoring the autocorrelation in motion energy profile. Thus, $e_{tr}^v$ is more accurately described as a measure of dispersion of the motion energy profile on single trials rather than as an estimate of the variance. The mean and the dispersion of the motion energy were used together with the stimulus duration ($T_d$) to train a logistic regression model to classify the volatility condition of each trial:

$$p_v^{tr} = \left[ 1 + e^{-\left( \beta_0 + \beta_1 |e_{tr}| + \beta_2 e_{tr}^v + \beta_3 T_d + \beta_4 |e_{tr}| e_{tr}^v \right)} \right]^{-1} \tag{24}$$

where $p_v^{tr}$ is the probability that trial $tr$ is of high volatility. After fitting the logistic model, we estimated the degree of overlap in the distributions of $p_v^{tr}$ between trials of low and high volatility. The area under the ROC curve was 0.895, indicating that there is information in the stimulus to reliably estimate the volatility condition of each trial, even for the brief stimulus presentations used in the PDW task. If we remove the interaction term ($\beta_4$) the area under the ROC curve is 0.85. To be clear, we do not put forward this calculation as a plausible model for inferring volatility. It merely serves to document that information is present in the stimuli to render a categorization possible.

## Analysis of neural data

For simplicity, in what follows we refer to both single units and multiunit sites as 'neurons'. To investigate how the volatility manipulation affected the mean and variance of the neuronal response, we first counted spikes occurring between 100 ms and 200 ms from stimulus onset. To avoid artifacts produced by the response to the offset of the RDM stimulus, we restricted this analysis to trials where the motion stimulus was presented for at least 150 ms. The counts were standardized (z-scored) independently for each neuron and subsequently grouped across neurons to obtain a large array of normalized counts, $s_{tr}$, where $tr$ indexes the trial number across sessions. *Figure 2C* shows the mean ($\mu_{c,v}$) and the variance ($\sigma_{c,v}^2$) of $s_{tr}$ computed over the subset of trials given by every combination of motion coherence and volatility condition.

These analyses furnished empirical estimates of the mean and variance of the spike count as a function of motion strength and direction. Findings from neurophysiology (*Ditterich et al., 2003*) and computational modeling (*Mazurek et al., 2003*) suggest that the momentary evidence is proportional to the difference of firing rates between pools of neurons with opposite direction preferences (e.g., right-preferring minus left-preferring). The expectation of this difference variable ($\Delta$) can be estimated empirically:

$$\mu_{\Delta|c,v} = \mu_{c,v} - \mu_{-c,v} \tag{25}$$

where $c$ and $-c$ indicate motion in the preferred and anti-preferred direction of the neuron, for motion strength $c$. The mean of the difference variable is shown in *Figure 2D*, with mean counts $\mu_{c,v}$ and $\mu_{-c,v}$ obtained from *Figure 2C*.

The variance of the difference variable ($\sigma_\Delta^2$) was approximated as follows. Because the variance of a sum equals the sum of the covariances, if the average pairwise correlation for a pool of $n$ neurons is given by $r$, then the variance of the average response of the pool is $\left( \frac{\sigma^2}{n} + \frac{n-1}{n} r\sigma^2 \right)$, where $\sigma^2$ is the variance in the spike counts from a single neuron. As $n$ becomes large (in practice, above 50 to 100 neurons is sufficient), the variance of the pool converges to $r\sigma^2$. Further, there is a portion of the variance that is shared between neurons tuned to the preferred and anti-preferred directions. If the correlation between the average responses of populations of neurons with opposite directional preferences is given by $\rho$, the variance of the difference variable as is given by *Equation 1* of the main text.

For the analyses depicted in *Figure 7*, we extracted the spike times from each trial up to 50 ms after motion offset and then smoothed the spike counts with a centered boxcar filter with a 30 ms width. For the analysis of *Figure 7B* we computed, for each neuron, the difference in firing rate between the response to the preferred and the non-preferred directions, for trials of the highest coherence (c = 0.512). This difference was used to estimate the latency with which motion information is represented in these neurons, regardless of the choice. For the analyses of *Figure 7A,C,D*, we obtained the residuals of firing rate by subtracting, from each trial and time step, the average firing rate of the same neuron on trials having the same motion direction, coherence and volatility. To group trials across neurons, we divided the activity of each neuron by a normalization constant, given by the maximum average firing rate at the highest coherence (i.e., c = 0.512). The latencies in *Figure 7B* were estimated with a curve fitting procedure based on the CUSUM method (*Ellaway, 1978*). In the CUSUM method, the latency of the difference between two conditions is estimated based the cumulative sum of the differences, thereby achieving robustness against the

noisiness of individual data point. The cumulative sum of differences was fit to a curve composed of two lines, the first of which was constrained to have a zero slope [similar to *Lorteije et al. (2015)*; *Van den Berg et al. (2016)*]. The latency is then estimated as the time point when the two lines intersect. Standard errors of the latency estimates were derived with a bootstrapping procedure (N = 1000).

## Acknowledgements

This research was supported by the Howard Hughes Medical Institute, the Human Frontier Science Program and the National Eye Institute (R01 EY11378). We thank Mariano Sigman, Luke Woloszyn and Daniel Wolpert for helpful discussions, and NaYoung So for comments on the manuscript.

## Additional information

### Funding

| Funder | Grant reference number | Author |
|---|---|---|
| Howard Hughes Medical Institute | | Ariel Zylberberg<br>Christopher R Fetsch<br>Michael N Shadlen |
| Human Frontier Science Program | | Michael N Shadlen |
| National Eye Institute | R01 EY11378 | Ariel Zylberberg<br>Christopher R Fetsch<br>Michael N Shadlen |

The funders had no role in study design, data collection and interpretation, or the decision to submit the work for publication.

### Author contributions

AZ, CRF, Conception and design, Acquisition of data, Analysis and interpretation of data, Drafting or revising the article; MNS, Conception and design, Analysis and interpretation of data, Drafting or revising the article

### Author ORCIDs

Ariel Zylberberg, http://orcid.org/0000-0002-2572-4748
Michael N Shadlen, http://orcid.org/0000-0002-2002-2210

### Ethics

Human subjects: The institutional review board of Columbia University (protocol #IRB-AAAL0658) approved the experimental protocol, and subjects gave written informed consent.
Animal experimentation: This study was performed in accordance with the recommendations in the Guide for the Care and Use of Laboratory Animals of the National Institutes of Health. All of the animals were handled according to approved institutional animal care and use committee (IACUC) protocols (AC-AAAE9004) of Columbia University.

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

## Appendix 1

# Derivation of the normative model

We used dynamic programming to determine how a rational decision-maker ought to adjust the height of the decision termination bounds when trials of different volatilities are randomly interleaved. For simplicity, we assume that the variance of momentary evidence is known to the decision maker – or that it can be estimated very rapidly (e.g., *Drugowitsch et al. [2014]*). As in previous studies (*Rao, 2010*; *Drugowitsch et al., 2012*; *Huang et al., 2012*) we derive the optimal strategy by representing the random-dot motion discrimination task as a partially-observable Markov Decision Process (POMDP). The solution to the POMDP is then derived by recasting it as an MDP (i.e., assuming full observability over the belief states) and using dynamic programming to derive the policy that maximizes average reward.

An MDP can be described as a tuple given by (*Bertsekas et al., 1995*; *Geffner and Bonet, 2013*):

i.   a non-empty state space $S$,

ii.  an initial state $S_0$,

iii. a goal state $S_G$,

iv.  a set of actions $A(s)$ applicable in state $s$,

v.   positive and negative rewards $r(a, s)$ for doing action $a$ in state $s$,

vi.  transition probabilities $Pa(s'|s)$ indexing the probability of transitioning to state $s'$ after doing action $a$ in state $s$.

The state $s$ was defined as a tuple $(x, t, v)$, where $x$ is the accumulated motion evidence for one direction and against the other (with its sign indicating the direction of motion), $t$ is elapsed time from the onset of motion, and $v$ is the volatility condition (low or high).

In the initial state, $x = 0$ (no net evidence favoring either of the alternatives), $t = 0$ and there is an equal probability of being in a high or low volatility regime.

Three actions are applicable in each state: two directional choices (e.g., left and right) and third action ('fix'), which is to maintain fixation for an extra time step to gather additional motion information. The outcome of the MDP is a deterministic policy, which assigns an action to each state.

Transition probabilities $P_a(s'|s)$ indicate the probability of transitioning to $s'$ after performing action $a$ in state $s$. As for the bounded accumulation model, the momentary motion evidence is assumed to be normally distributed with a mean that depends linearly on motion coherence ($c$), and variance $\sigma_v^2 \delta t$. After $t$ sec, the accumulated evidence would—in the absence of bounds—also be normally distributed with mean $t\kappa c$ and variance $t\sigma_v^2$. Here we assume that $\sigma_v^2$ is independent of coherence to avoid additional complexities in the numerical solution of Bellman's equation. Note that this simplification departs from the volatility manipulation introduced in the experiment.

For a given motion coherence, the probability that the evidence gathered in a time step $\delta t$ leads to a transition from state $s = (x, t, v)$ to state $s' = (x', t + \delta t, v)$ is given by:

$$p_{fix}(s'|s, c) = N\left(x' - x | \kappa.c.\delta t, \sigma_v \sqrt{\delta t}\right) \tag{A1}$$

where $N(\cdot|\mu, \sigma)$ is the normal p.d.f. with mean $\mu$ and standard deviation $\sigma$.

We then need to marginalize over coherences to obtain the transition probability $p_{fix}(s'|s)$:

$$p_{fix}(s'|s) = \sum_c p_{fix}(s'|s,c)p(c|s) \tag{A2}$$

Marginalizing over coherences requires knowledge of $p(c|s)$, the probability that the motion coherence is $c$, given that state $s$ was reached, which can be computed as:

$$p(c|s) = p(c|x,t,v) \propto N(x|\kappa.c.t, \sigma_v\sqrt{t})p(c) \tag{A3}$$

where the coherences $c$ are the discrete set of signed coherences used in the experiment, and the proportionality constant is such that that the sum of $p(c|x,t,v)$ over all motion coherences adds to one (*Moreno-Bote, 2010*). As in the experiment, $p(c)$ is distributed uniformly over the discrete set of unsigned motion coherences.

The policy that maximizes average reward was found using value iteration to numerically solve Bellman's equation. The process works by assigning to every state, $s$, a value $V(s)$, which is the largest associated with the three possible actions: choose right ($r$), choose left ($l$), or continue gathering evidence (*fix*):

$$V(s) = max \begin{cases} Q(s,r) & = b(s,r)R_c + (1-b(s,r))(R_{nc} - t_p\rho) - (t_{nd} + t_w)\rho \\ Q(s,l) & = b(s,l)R_c + (1-b(s,l))(R_{nc} - t_p\rho) - (t_{nd} + t_w)\rho \\ Q(s,fix) & = \int_{s'\in S} ds' p_{fix}(s'|s)V(s') - \rho\delta t \end{cases} \tag{A4}$$

where $b(s,a)$ is the probability of being correct after doing action $a$ in state $s$; and $R_{nc}$ are the rewards following correct and incorrect decisions (here 1 and 0 respectively); $t_p$ is the time penalty after an error, $t_{nd}$ is the average non-decision time, and $t_w$ is the average time spend between decisions including the time spend acquiring fixation and observing feedback; $\rho$ is the amount of reward obtained per unit of time (explained further below).

The probability of being correct after doing action $a$ in state $s$, $b(s,a)$, can be obtained summing over the coherences for which the action $a$ is the appropriate action. For instance, the action 'right' is the appropriate action for all positive coherences and for half of the 0% coherence trials. Therefore,

$$b(s,r) = b((x,t,v),r) = \sum_{c>0} p(c|x,t,v) + \frac{1}{2}p(c=0|x,t,v) \tag{A5}$$

Because choosing right is a terminating event, there is no need to consider future states, and the same applies to the left choice. The value of gathering additional evidence before committing to a choice is captured by $Q(s,fix)$, computed as an expectation over all future states $s'$ that result from being in $s$ and gathering evidence for an additional time step $\delta t$.

Because time flows in a single direction, if the reward rate were known, then Bellman's equation can be solved by backwards induction in a single pass. Since the reward rate depends on the policy itself, we perform multiple backward passes, bracketing $\rho$ within a sequence of diminishing intervals until the value of the initial state $V(S_0)$ becomes vanishingly small (*Bertsekas et al., 1995*; *Drugowitsch et al., 2012*). The procedure yields a formulation of the stopping criteria as a function of time. These are the optimal bounds shown in *Appendix 1—figure 1* (top row) for different scenarios (*Appendix 1—table 1*).

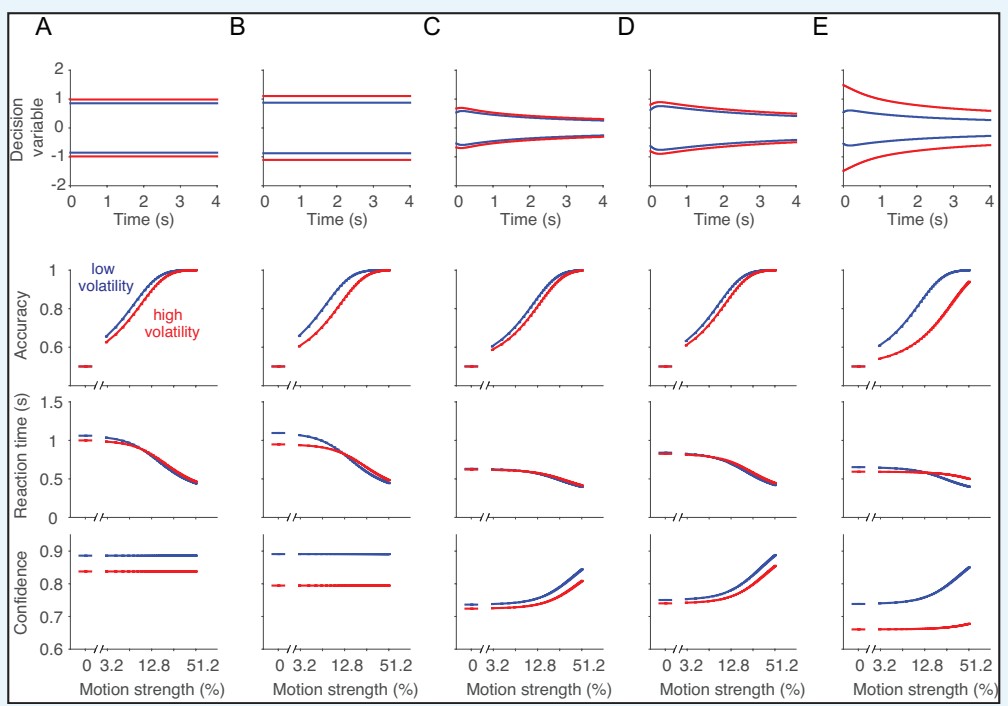

**Appendix 1—figure 1.** Normative decision model. From the model comparisons we concluded that the volatility manipulation affected the noisiness of the momentary evidence without consistently affecting the parameters of the decision making process. However, it is unclear from these analyses what an optimal decision maker would do if the volatility condition were known. Therefore, we used dynamic programming to find the decision policy that maximizes average reward (**Rao, 2010**; **Drugowitsch et al., 2012**), for a variety of combinations of task parameters. The following examples are meant to convey intuitions about how parameters can change to maximize overall success per unit time. (**A**) Optimal solution for an experiment in which there is just one nonzero motion strength. Notice that the optimal bound height for low and high volatility trials is independent of time, consistent with a well known property of Wald's sequential probability ratio test (**Wald and Wolfowitz, 1948**). The high volatility condition invites a slightly higher bound but not so much to overcome the faster decision times induced by greater noise (**A**, third row). Unlike the experimental observation, the normative solution assigns lower confidence under high volatility (**A**, bottom). (**B**) If the noise level associated with the high volatility condition were exaggerated further, the optimal solution would predict a greater increase in the bound height, thereby compensating for the additional noise. The bound height for the low volatility condition should increase as well. (**C**) In the situation we study, there are many levels of difficulty which are randomly interleaved across trials. In this situation, the optimal solution asserts a time-dependent collapse of the bounds toward lower magnitude of accumulated evidence (**Drugowitsch et al., 2012**). As in the single coherence case, the high volatility condition should induce an increase in bound height at all times, relative to the low volatility condition. Notice, however, that the optimal solution would lead to lower confidence under high volatility—contrary to what we observed empirically. The same pattern holds if there is a substantial time penalty after an error (**D**) and if the variance in the high volatility condition were exaggerated to six times that of the low volatility condition (**E**).

With the optimal bounds for each volatility condition, we compute the probability that the decision was correct given that a bound was reached at time $t$. For a single motion coherence (**Appendix 1—figure 1A–B**), this probability is independent of time (**Wald and Wolfowitz, 1948**). For different sets of parameters (**Appendix 1—table 1**) we derive the choice and decision time by solving numerically the Fokker-Planck equations using the optimal bounds,

for a fine grid of coherence values and for both volatility conditions (second and third rows of the figure). The confidence for correct choices (*Appendix 1—figure 1*, bottom row) in this model is determined solely by the time [see *Kiani et al. (2014)*]. The confidence associated with each coherence was obtained by marginalizing the probability correct at the bound over the distribution of decision times obtained for each motion coherence and volatility condition.

**Appendix 1—table 1.** Parameters explored in the normative model.

| Panel in *Appendix 1—figure 1* | A | B | C | D | E |
|---|---|---|---|---|---|
| coherence set | ± 0.1 | ± 0.1 | as in the experiments | as in the experiments | as in the experiments |
| $\kappa$ | 12 | 12 | 12 | 12 | 12 |
| $\sigma^{low}$ | 1 | 1 | 1 | 1 | 1 |
| $\sigma^{high}$ | 1.2 | 1.4 | 1.2 | 1.2 | 2.5 |
| $t_{nd}$ (s) | 0.3 | 0.3 | 0.3 | 0.3 | 0.3 |
| $t_p$ (s) | 0 | 0 | 0 | 2 | 0 |
| $t_w$ (s) | 3 | 3 | 3 | 3 | 3 |

While none of the normative models depicted in *Appendix 1—figure 1* correspond in detail to the experiment we conducted, the analysis carries three implications which are likely to apply. First, if the volatility conditions (low or high) were known, the decision maker should adjust the termination criteria and confidence mapping. In other words, it would be desirable to know the volatility conditions and to adjust the decision process accordingly. Second, if subjects approximated the optimal behavior they would have been less confident on high volatility trials. The observation that they were more confident on these trials implies that they were not optimal, or they could not identify the volatility without adding additional costs (e.g., effort and/or time). Third, the faster RT on high volatility trials would have been expected even if the subjects had applied different decision criteria.

