## [Decision Letter]

Thank you for submitting your article "The influence of evidence volatility on choice, reaction time and confidence in a perceptual decision" for consideration by *eLife*. Your article has been reviewed by three peer reviewers, and the evaluation has been overseen by a Reviewing Editor and David Van Essen as the Senior Editor. The following individuals involved in review of your submission have agreed to reveal their identity: Hakwan Lau (Reviewer #1); Jeff Beck (Reviewer #3).

The reviewers have discussed the reviews with one another and the Reviewing Editor has drafted this decision to help you prepare a revised submission.

Summary:

This article reports an experiment using a random dot motion stimulus in which the coherence is a random variable from frame-to-frame with a given mean and standard deviation. The experiment manipulated the mean coherence and the volatility of the coherence with either no frame-to-frame variance (low volatility) or with frame-to-frame variance (high volatility). Choice, response time, and measures of retrospective confidence were collected from three human participants and one macaque monkey. For the monkey, confidence monkey was measured with post-decision wagering. For the humans, confidence was measured via a scale positioned on the left and right of the stimulus allowing a choice to be recorded at the same time a confidence level was recorded. Mean coherence had the typical effects on all 3 measures. Volatility, particularly at the lower levels of coherence, decreased response time and increased the average level of confidence. The results from detailed analyses using a motion energy analysis of the stimulus, firing rate of direction selective neurons in the MT, and behavioral data, are reported as consistent with a bounded evidence accumulation model where sensory information is accumulated as evidence until it reaches a threshold or stimulus offset. This determines the observed choice and response time. Confidence is assumed to reflect the probability the decision is correct given the level of evidence reached, though an important conditional here is that this mapping is not impacted by the so-called volatility of the stimulus.

Essential revisions:

While all reviewers were enthusiastic about the potential contribution, noting that it was timely and methodologically rigorous, they also raised some substantive issues that need to be addressed with further analysis. Individual reviewer comments are amalgamated below. You will note that some of the comments are redundant. We have kept the redundancy to reflect both the overlapping views, but also the different means that might be taken to address the concerns.

1) A major concern with the paper echoed by multiple reviewers (see related points 2-4 below) and reinforced in discussion is that for deriving confidence judgments there is a strong assumption that the decision maker does not adjust the mapping between the location of the DV and probability of a correct choice. Is this an assumption that the authors had before seeing the data, which seems hard to defend? If so it might be worth a few sentences defending why they assumed this. If not this is also fine, but this should also be made explicit.

2) Strong assumption of variability. There is a pretty strong assumption in the model that to account for the confidence data that "the subject has implicit knowledge of the mapping between the DV and probability correct, and does not adjust the mapping when a high volatility stimulus is shown" (–Results). It seems, at first glance, consistent with the data, but inconsistent with the motivation behind the model itself. That is, how can confidence reflect the probability the decision is correct given the decision variable, but then that mapping is invariant to manipulations that in principle impacts the probability of being correct? Choice accuracy was not significantly impacted, but the trends were certainly in the correct direction with volatility leading a small drop in accuracy (see Figure 3). More justification for this assumption is needed. Theoretically this just doesn't seem to be consistent with the a priori principles of the model because in principle more variance should impact accuracy. Isn't it also the case that this assumption implies there should be no change between low and high volatility conditions in the average difference in the confidence given to correct vs incorrect choices? I suspect the difference in confidence between correct and incorrect choices (at least for the human data) is smaller in the high volatility condition.

One way to address this is with fitting an ideal observer model. Another way (which might lead to the same model) is to fit the same model but allowing the mapping to change and compare goodness of fits of the two models. It might just be that this data lead to this conclusion. I would also be curious to what degree confidence in corrects and confidence in incorrect is different, at least in humans. I think their model with the invariance assumption predicts that there should be no change between variability conditions in this difference score. I believe it would be informative to see this examined.

3) Methods: "We derived the mapping using the variance from the low volatility condition. We assume, however, that the same mapping supports confidence ratings (and PDW) in both volatility conditions." This is a problematic assumption at best and really needs a more thorough justification. It's tantamount to the assumption that the system is not estimating volatility on a trial by trials basis as an ideal observer would when high and low viability trials are interleaved. As a result, it is unclear whether or not these results are a product of this assumption or a product of the task itself. We know from human psychophysics that we do estimate volatility (or stimulus variance) on a trial by trial basis and more or less rationally incorporate that information into our decision variables. In this context an ideal observer model should be associated with an accumulator that rises sub-linearly it accumulates evidence that this is a high volatility trial. While the opposite is true in low volatility trials. If there is not enough information to accurately estimate stimulus variability then the author's assumption is valid. But it's not clear that this is the case. Moreover, even an ideal observer model may exhibit the same qualitative behavior as the author's suboptimal model even when there is enough information to estimate variability accurately. It would be nice to know if this is the case, but regardless, some discussion of these issues would be a welcome addition to this work.

4) Moreover, if I am right about how you set your prior on c then it shouldn't be too hard to adapt the code you have already written to just make the sum in Equation 5 include the two volatility conditions and then just talk log and call this your decision variable. This decision variable will not be a linear sum of the momentary evidence. It will be sublinear when volatility is high and superlinear with volatility is low. This actually fits very nicely with your neural data which shows that in the high volatility condition the anti-preferred neurons slightly increase in activity.

5) On this point about fixed boundary, how important it is that the high and low volatility conditions are mixed? Would it have worked if they were blocked in different days? Weeks? I.e. how inflexible are the subjects in shifting from one set of bound to another? Finding it in monkeys may be too much work, but can we do this in human subjects? Seems most plausibly, that in normal everyday circumstances, adding noise should make people slower and less confident, not the other way round. So these effects may be limited to the experimentally contrived conditions where subjects are overtrained on trials with different levels of noise mixed randomly.

6) A related but perhaps broader point, regarding confidence. If subjects adopt a fixed set of bounds/criteria, their decision mechanism seems decidedly suboptimal/heuristical. However, many prominent researchers now advocate the approach of first writing down the optimal mathematical definition of confidence, and then proceed to find its correlates in the brain (cf Kepecs, Pouget, etc.). Given that confidence can be assessed in animals including rodents, an alternative approach is to empirically assess confidence which may not be optimal. In this context, don't the present findings give important and stern warning to the "optimality" approach? This is a key issue that may influence the agenda of the field for years to come, and should be emphasized.

7) The neuronal recordings also add to the novelty beyond previous work, but much more details are needed – right now it's almost like an afterthought. As noted by the authors, the basic conclusion that adding noise leads to increased confidence has already been shown. Can we plot the fano factor of these neurons at different conditions? The impact of stimulus noise seems modest, but does it change pair-wise noise correlation between neurons too? Even if it doesn't, what is the correlation to begin with, for these neurons? Since multi-unit recording was performed in at least some of these neurons, we should have some idea? If pairwise noise correlation is high, it limits the efficiency of a readout mechanism (if such mechanism is to do anything resembling averaging of individual neuronal responses), and may thus mean that the readout is noisy too (because individual noise can't be efficiently averaged out). Again, I understand there are relatively few neurons here, but this is an important part of the results, going beyond previous studies, and should perhaps be mentioned in the Abstract, so people will know this is not *just* a psychophysics paper.

8) I had a hard time appreciating whether the more extreme confidence judgments were diagnostic of this particular model of choice, response time, and confidence, or if other models would also predict this result. For instance, Pleskac and Busemeyer's 2DSD model (assuming say a serial process of choice then confidence) would also predict higher average confidence at lower levels, but for a slightly different reason with the variability combined with a bounded scale would produce regressive like effects. It may be the case the monkey data with post-decision wagering would speak against this, but it seems like a relevant discussion item.

Pleskac, T. J., & Busemeyer, J. R. (2010). Two-Stage Dynamic Signal Detection: A Theory of Choice, Decision Time, and Confidence. Psychological Review, 117, 864-901. doi:10.1037/A0019737

---

## [Author Response]

*Essential revisions:*

*While all reviewers were enthusiastic about the potential contribution, noting that it was timely and methodologically rigorous, they also raised some substantive issues that need to be addressed with further analysis. Individual reviewer comments are amalgamated below. You will note that some of the comments are redundant. We have kept the redundancy to reflect both the overlapping views, but also the different means that might be taken to address the concerns.*

We thank the reviewers for their comments and the reviewing editor for compiling them. We have addressed all of the concerns, both in this response letter and in the revised manuscript which contains several new analyses and discussion points. We first describe our response to the major concern described in section 1 below, then touch on any additional aspects of this concern as they come up in subsequent comments.

*1) A major concern with the paper echoed by multiple reviewers (see related points 2-4 below) and reinforced in discussion is that for deriving confidence judgments there is a strong assumption that the decision maker does not adjust the mapping between the location of the DV and probability of a correct choice. Is this an assumption that the authors had before seeing the data, which seems hard to defend? If so it might be worth a few sentences defending why they assumed this. If not this is also fine, but this should also be made explicit.*

We assumed a fixed mapping between the DV and confidence because the volatility manipulation is fairly subtle (see Video 1) and the trial types were interleaved and uncued. Indeed, subjects were not told and had no reason to suspect that anything other than motion strength was being varied across trials, let alone that there were exactly two volatility conditions. Keep in mind that motion strength itself is a type of reliability, and we know from many lines of evidence that subjects do not immediately identify the motion strength in this task. For example, if they did, RT would be shortest for trials of 0% coherence because reward probability is 0.5 by definition, thus the sensible thing to do is guess quickly.

We also note that the primary goal of the study was not to evaluate the assumption of a fixed mapping, but to test a more fundamental prediction of the bounded evidence accumulation framework, namely the effect of manipulating noise independent of signal on choice, RT and confidence.

Nevertheless, we agree that the issue is important and deserves additional attention in the paper. To this effect, we have added the following analyses/figures:

A) We tested whether there is sufficient information in the stimulus to discriminate between volatility conditions. We did this by training a logistic regression model which had access to the mean and dispersion of motion energy (and their interaction) on each trial of the PDW task (subsection “Alternative models”, first paragraph; subsection “Motion Energy”, last paragraph). The logistic model was able to discriminate volatility conditions with 90% accuracy, suggesting that it is reasonable to ask whether subjects did in fact use this information to adjust their decision policy.

B) To determine whether they did, we fit three alternative models in which the volatility condition affects the mapping between DV and probability correct. These are explained in a new subsection of Results (“Alternative Models”, first paragraph). The first version simply assigns separate mappings to the two volatility conditions (derived from the variance of the momentary evidence in each condition). This is tantamount to assuming that subjects identify the volatility instantaneously. The second version permits volatility to be estimated gradually, and the third allows the termination criterion (bound height) to differ as well. A Bayesian model comparison showed that none of the alternative models were as well supported by the data as the commonmapping model (see Table 2 and new Figure 6).

C) We then went a step further and asked what an ideal observer would do given knowledge of the volatility of each trial. For instance, if errors are inexpensive it is conceivable that the best strategy in the high volatility condition is to respond rapidly, to hasten onset of the next trial. However, if errors are costly, for instance if they lead to a large timeout penalty, then the rational strategy for high volatility stimuli could be to increase the bound height to maintain high levels of accuracy even under high volatility. We used dynamic programming to derive the optimal policy (in the sense of maximizing reward rate) when the volatility of the evidence can differ between trials (see Appendix).

As described in detail in the new Appendix, we found that the optimal policy includes an increase in the height of the termination bound (Figure 8, top row), as well as the aforementioned change in the mapping between DV and probability correct. Interestingly, the added bound height is not enough to overcome the effect of noise on RT, such that an optimal decision maker would still show slightly faster RTs at low coherence in the high volatility condition (Figure 8, third row). However, the normative solution predicts lower confidence on highvolatility trials (bottom row), which we did not observe in our subjects. Again, as our main focus was not on the optimality (or lack thereof) of our subjects in this task, we situated the analysis of normative models in an appendix.

D) Lastly, given the assumption of a common mapping, there is the question of which mapping to use. In our original submission, we assumed that the mapping used for both volatility conditions was constructed based on the variance of the lowvolatility condition. The motivation for this assumption was that subjects, especially the monkeys, had extensive experience with the low volatility condition before we introduced the high volatility condition. Given the subtlety of the manipulation, and the lack of a strong effect on accuracy, subjects may have had little impetus to conclude that task contingencies have changed when we introduced the volatility manipulation, and would therefore continue to rely on the mapping that they learned for the low volatility condition. Based on the reviewer’s comments, we now realize that this reasoning was unclear, as we did not specify the conditions under which the mapping is or is not expected to change, and how much experience is required for the mapping to change. Therefore, we now redefine the mapping to include both volatility conditions. The map was derived rationally, meaning that it was obtained marginalizing over the two volatility conditions, as it is explained in Results (subsection “Effects of volatility on confidence”, first paragraph) and Methods (new equation 5). We believe this to be more parsimonious and more consistent with the extensive experience that our subjects eventually developed under the mixture of volatilities. We note, however, that the two models produce almost identical fits. The reason for this is better understood with Figure 9, which is equivalent to the new Figure 6 but for three maps (instead of two): low volatility (dashed lines), high volatility (dotted lines) and mixture of high and low volatilities (solid lines). As can be observed in the maps, the isoprobability contours are approximately scaled versions of one another. Because the criterion Φ that separates high from low confidence is a parameter that we fit to data, we cannot distinguish between a low volatility map with a Φ of (say) 0.6 from a mixture map with a Φ of ~0.56.

Author response image 1.Iso-confidence contours for the mappings derived from the low volatility condition, the high volatility condition, and the mixture of low and high volatilities.The figure is similar to the new main Figure 6, except that it also shows the map derived from the mixture of volatilities.**DOI:**
http://dx.doi.org/10.7554/eLife.17688.022

*2) Strong assumption of variability. There is a pretty strong assumption in the model that to account for the confidence data that "the subject has implicit knowledge of the mapping between the DV and probability correct, and does not adjust the mapping when a high volatility stimulus is shown" (–Results). It seems, at first glance, consistent with the data, but inconsistent with the motivation behind the model itself. That is, how can confidence reflect the probability the decision is correct given the decision variable, but then that mapping is invariant to manipulations that in principle impacts the probability of being correct? Choice accuracy was not significantly impacted, but the trends were certainly in the correct direction with volatility leading a small drop in accuracy (see Figure 3). More justification for this assumption is needed. Theoretically this just doesn't seem to be consistent with the a priori principles of the model because in principle more variance should impact accuracy.*

We appreciate the conundrum raised by the reviewer, and thank him/her for the opportunity to make an important point. First, any mechanism or criterion for establishing confidence in a perceptual decision must be invariant to at least one manipulation that affects probability correct, namely the difficulty or stimulus strength itself. If the brain were able to identify perfectly the difficulty level (motion coherence in our tasks), it would have no need to estimate probability correct from a DV, but would simply assign a level of confidence based on the actual probability correct experienced at that difficulty. In the extreme case, decisions on 0% coherence trials should all be rendered with the lowest possible confidence level (regardless of the DV), but this is not what happens. Volatility is simply another parameter of the visual stimulus that, as we have now shown, does not appear to affect the mapping of a DV to confidence (leaving aside the empirical observation that our volatility manipulation had very weak effects on accuracy, providing little incentive for the brain to adjust to it).

In short, the idea that confidence reflects an estimate of probability correct is not at all inconsistent – logically or empirically – with the mechanism being invariant to a stimulus manipulation that affects accuracy, especially one that is subtle and randomly interleaved. Nevertheless, the reviewer’s general concern about the assumption of an invariant mapping is valid, and we hope we have addressed it in the response to item #1 above.

*Isn't it also the case that this assumption implies there should be no change between low and high volatility conditions in the average difference in the confidence given to correct vs incorrect choices? I suspect the difference in confidence between correct and incorrect choices (at least for the human data) is smaller in the high volatility condition.*

We do not completely follow the reviewer’s intuition, but we performed the suggested analysis and found no clear indication for a greater difference in confidence – between correct and incorrect choices – for the low volatility condition. For each subject, coherence, and volatility condition, we computed the difference in confidence (saccadic endpoint) between correct and error trials. Figure 10 shows the result of this analysis. Colors indicate different coherences, and shapes denote different subjects. We found no strong evidence for differences in confidence between correct and error trials among the two volatility conditions. We note, however, that there are many caveats to this analysis that the current dataset cannot resolve. First, the analysis would require a larger number of trials to obtain a more reliable estimate of confidence on error trials, especially for the higher coherences. Second, the subjects reported confidence on an uncalibrated confidence scale, therefore it is difficult to map differences in this arbitrary scale to differences in subjective probabilities or degrees of belief. Given these caveats, we would rather not include this figure in the manuscript.

Author response image 2.Difference in confidence (saccadic endpoint) between correct and error trials for high vs. low volatility.**DOI:**
http://dx.doi.org/10.7554/eLife.17688.023

*One way to address this is with fitting an ideal observer model. Another way (which might lead to the same model) is to fit the same model but allowing the mapping to change and compare goodness of fits of the two models. It might just be that this data lead to this conclusion. I would also be curious to what degree confidence in corrects and confidence in incorrect is different, at least in humans. I think their model with the invariance assumption predicts that there should be no change between variability conditions in this difference score. I believe it would be informative to see this examined.*

We thank the reviewer for this very useful suggestion. We have now performed both analyses suggested by the reviewer: (1) fit several alternative models where the mapping is allowed to change (new Figure 6), and (2) derive an ideal observer model for our task (see new Appendix). See response to item #1 above for details. Regarding the issue of comparing the differences in confidence between correct and errors for both volatilities, please see the response to the previous point.

*3) Methods: "We derived the mapping using the variance from the low volatility condition. We assume, however, that the same mapping supports confidence ratings (and PDW) in both volatility conditions." This is a problematic assumption at best and really needs a more thorough justification. It's tantamount to the assumption that the system is not estimating volatility on a trial by trials basis as an ideal observer would when high and low viability trials are interleaved. As a result, it is unclear whether or not these results are a product of this assumption or a product of the task itself. We know from human psychophysics that we do estimate volatility (or stimulus variance) on a trial by trial basis and more or less rationally incorporate that information into our decision variables. In this context an ideal observer model should be associated with an accumulator that rises sub-linearly it accumulates evidence that this is a high volatility trial. While the opposite is true in low volatility trials. If there is not enough information to accurately estimate stimulus variability then the author's assumption is valid. But it's not clear that this is the case. Moreover, even an ideal observer model may exhibit the same qualitative behavior as the author's suboptimal model even when there is enough information to estimate variability accurately. It would be nice to know if this is the case, but regardless, some discussion of these issues would be a welcome addition to this work.*

The concern raised by the reviewer’s quote of our text is that we used the variance from the low volatility condition to derive the mapping used in both conditions. Above, under item 1D, we explain our reasoning for doing this in the original submission, and that we have now changed the model to reflect a mixture of the two volatility conditions. However, we suspect this issue is secondary to the broader questions surrounding whether a common mapping is used at all: (i) whether there is enough information to estimate stimulus variability on a trial by trial basis (subsection “Alternative models”, first paragraph; subsection “Motion Energy”, last paragraph), (ii) whether our subjects exploited such information (subsection “Alternative Models*”*), and (iii) what an ideal observer ought to do (Appendix). We hope these aspects of the reviewer’s concern are fully addressed in our responses to item #1 and in the revised manuscript.

Regarding the point that “we know from human psychophysics that we do estimate volatility (or stimulus variance) on a trial by trial basis[…]”, we agree with the reviewer that there are situations where subjects do take into account the reliability of the evidence when making a decision. When reliability is easily discernible—as when the contrast of the stimulus is markedly reduced—observers may adjust their decision criteria in a rational manner.

However, our task is representative of a class of problems in which reliability is neither explicitly cued nor easily discernible. Under such circumstances, whether subjects would still try to infer the reliability of the evidence is an open question. Our data are consistent with the interpretation that extracting information about volatility from a stream of momentary evidence and using it to adjust the parameters of the decision process (e.g., the bound height or the mapping of DV to probability correct) is not an automatic process. This, however, does not imply that this volatility information will be ignored if more easily available or if ignoring volatility has more severe consequences, as we address in Discussion (fourth paragraph).

Regarding the point about sub and supralinear accumulation, we are aware that this is required to obtain Bayesian-optimal solutions for decisions when the reliability of the evidence changes dynamically within a trial. Our situation is different because the noisiness in the momentary evidence is constant within a trial, and therefore scaling the evidence by the noise would be equivalent to a change in bound height. It is conceivable that subjects could have estimated the volatility of the evidence online in order to adjust the weight of the evidence, but we found no support for this hypothesis, as discussed in the new section on Alternative Models.

The new analysis of the neural data – which we added to address another suggestion by the reviewers – also argues against this interpretation. If there is a different adjustment of the weights of the evidence as a function of time for trials of low and high volatility, then the covariation between the neural responses and behavior should have different time courses for trials of low and high volatility. For instance, in the low volatility condition, late evidence should be weighted more because the monkey would have more certainty about the trial’s volatility condition after observing many samples of evidence. This prediction is not supported by the data. We found that the temporal profiles of the choice-conditioned firing rate residuals were similar for low and high volatilities, as can be seen in the new Figure 7.

*4) Moreover, if I am right about how you set your prior on c then it shouldn't be too hard to adapt the code you have already written to just make the sum in Equation 5 include the two volatility conditions and then just talk log and call this your decision variable. This decision variable will not be a linear sum of the momentary evidence. It will be sublinear when volatility is high and superlinear with volatility is low. This actually fits very nicely with your neural data which shows that in the high volatility condition the anti-preferred neurons slightly increase in activity.*

We hope we have clarified this issue above. Equation 5 has now been changed. Before, it conveyed the dependency of confidence (probability correct) on DV and time for the low volatility condition. In the revised manuscript the dependency reflects both volatility conditions – that is, a mixture with equal weighting from both, as the reviewer suggests.

Regarding the sublinear/superlinear weighting of the momentary evidence to construct the DV, we believe the reviewer is suggesting a model in which the decision maker (or brain) estimates the volatility of the momentary evidence onthefly. As indicated in the previous response, we now consider a related model in which the information about the volatility condition develops gradually during the trial, and found no support for the proposal. The exploration of normative models confirms the reviewers’ intuition that it would be ideal to ascertain volatility on the fly and to adjust policy accordingly, but the same normative models predict that confidence would be reported as lower under high volatility, which is not the case in our data.

Regarding the slight increase in activity for the anti-preferred direction in the high volatility condition, this is explained by the rectification of the input-output response of MT/MST neurons. In the high volatility condition, for the lowest coherences the motion can reverse direction in some frames. Therefore, even if the manipulation is symmetric in terms of the dot displacement, it is not symmetric when pushed through the nonlinear input-output response of real neurons (as shown in the inset of Figure 2).

*5) On this point about fixed boundary, how important it is that the high and low volatility conditions are mixed? Would it have worked if they were blocked in different days? Weeks? I.e. how inflexible are the subjects in shifting from one set of bound to another? Finding it in monkeys may be too much work, but can we do this in human subjects? Seems most plausibly, that in normal everyday circumstances, adding noise should make people slower and less confident, not the other way round. So these effects may be limited to the experimentally contrived conditions where subjects are overtrained on trials with different levels of noise mixed randomly.*

We do not know how subjects would have responded if high and low volatility conditions were blocked over days or weeks. However, it is interesting to note that even the alternative model that presumes knowledge of volatility (and the appropriate mapping) predicts higher confidence at the weak motion strengths (notice that the red curve in Figure 6 is above the blue; ignore the misses in the fits). This is because the increased noise in the high-volatility condition makes it more probable that the DV will reach the space normally occupied by the higher coherences. The same intuition explains our observation that, despite increasing the bound height on high volatility trials, an ideal observer still makes faster decisions on those trials, at least when motion strength is weak (see Figure 8, 3rd row). There certainly may be circumstances where a noise manipulation leads to slower and less confident decisions, in accordance with the reviewer’s intuition. But it is not obvious why such circumstances should be deemed more natural or general than the conditions of our experiment, in which reliability can vary unpredictably.

*6) A related but perhaps broader point, regarding confidence. If subjects adopt a fixed set of bounds/criteria, their decision mechanism seems decidedly suboptimal/heuristical. However, many prominent researchers now advocate the approach of first writing down the optimal mathematical definition of confidence, and then proceed to find its correlates in the brain (cf Kepecs, Pouget, etc.). Given that confidence can be assessed in animals including rodents, an alternative approach is to empirically assess confidence which may not be optimal. In this context, don't the present findings give important and stern warning to the "optimality" approach? This is a key issue that may influence the agenda of the field for years to come, and should be emphasized.*

The reviewer raises an important point, bearing on a larger set of ideas, but we are not keen on editorializing in this paper. While our results suggest that participants did not use information about volatility to adjust the DV-confidence mapping or the termination bounds, we cannot claim that this is necessarily suboptimal. To make this claim, we would need to know the cost in time and effort of estimating the volatility online to use it to adjust the parameters of the decision making process. So we would prefer to remain uncommitted on the optimality issue. We now mention this point explicitly in the Results section: "[…] we do not know if our subjects performed suboptimally or if they were simply unable to identify the volatility conditions without adding additional costs (e.g., effort and/or time). "

*7) The neuronal recordings also add to the novelty beyond previous work, but much more details are needed – right now it's almost like an afterthought. As noted by the authors, the basic conclusion that adding noise leads to increased confidence has already been shown. Can we plot the fano factor of these neurons at different conditions? The impact of stimulus noise seems modest, but does it change pair-wise noise correlation between neurons too? Even if it doesn't, what is the correlation to begin with, for these neurons? Since multi-unit recording was performed in at least some of these neurons, we should have some idea? If pairwise noise correlation is high, it limits the efficiency of a readout mechanism (if such mechanism is to do anything resembling averaging of individual neuronal responses), and may thus mean that the readout is noisy too (because individual noise can't be efficiently averaged out). Again, I understand there are relatively few neurons here, but this is an important part of the results, going beyond previous studies, and should perhaps be mentioned in the Abstract, so people will know this is not just a psychophysics paper.*

We thank the reviewer for this suggestion. As we now discuss in the text (subsection “Choice- and confidence-predictive fluctuations in MT/MST activity”, first paragraph), the main goal of the physiological recordings was to verify that the manipulation had the desired effect on the mean and the variance of the firing rate of neurons which presumably represent the momentary evidence in this task. We now mention this finding in the Abstract. The result of these analyses were summarized in Figure 2, an important prerequisite for the psychophysics and modeling that form the bulk of the paper. It was not an afterthought, but the data were acquired with this purpose and powered appropriately. That said, we agree with the reviewer that the data deserve further analysis. Accordingly, we added a short section in Results and a new figure that displays the time course over which the neural response informs confidence as assayed in the PDW task. Previous studies have shown correlations between the fluctuations of neural activity in MT/MST and directional choices in tasks similar to the one studied here. We now show that fluctuations in neural activity are also informative of whether the monkey will waive the sure target if available (new Figure 7). The spikes are informative about PDW at the same time that they are informative about motion direction and choice. The observation is consistent with the idea that choice and confidence are influenced by the earliest samples of motion information.

As suggested by the reviewer, we computed the Fano factor for the different conditions of motion coherence and volatility. For each neuron, we compute the mean and variance of the spike counts in a 100 ms counting window (100-200 ms after motion onset). For this analysis, we only included single neurons and trials in which the motion stimulus was presented for at least 150 ms. We then compute the variance-to-mean ratio for each neuron. The means ( ± SEM) are displayed in Figure 11. It shows that (i) the average Fano factor was above 1 for both volatility conditions and for all coherences, (ii) the Fano factor tends to increase with the strength of motion in the preferred direction, and (iii) there is an influence of volatility on the Fano factor, which is on average greater when the volatility is high.

We believe, however, that this analysis does not add much to the conclusions that can be derived from the analysis of main Figure 2, and therefore we prefer not to include it in the main text. Further, while the Fano factor is usually considered to be a characterization of the internal noise, here it includes contributions from both the internal noise and the fluctuations in the stimulus. Indeed, the measured variancetomean ratio is probably never a proper characterization of purely internal or external noise, or spike generation given a latent rate (see Shadlen & Newsome, 1998; Churchland et al., 2011; Nawrot et al., 2008), which is why Fano factor is a misnomer as it is commonly applied. Therefore, the increase in “Fano factor” with the motion coherence and with the volatility of the stimulus may mainly (but not entirely) reflect that the motion stimulus had higher variance in these conditions.

Author response image 3.Variance-to-mean ratio of spike counts is greater under high volatility.Trials of low and high volatility are shown in blue and red, respectively. Positive motion coherences correspond to motion in the neuron’s preferred direction. Points are averages over N=26 single neurons.**DOI:**
http://dx.doi.org/10.7554/eLife.17688.024

*8) I had a hard time appreciating whether the more extreme confidence judgments were diagnostic of this particular model of choice, response time, and confidence, or if other models would also predict this result. For instance, Pleskac and Busemeyer's 2DSD model (assuming say a serial process of choice then confidence) would also predict higher average confidence at lower levels, but for a slightly different reason with the variability combined with a bounded scale would produce regressive like effects. It may be the case the monkey data with post-decision wagering would speak against this, but it seems like a relevant discussion item.*

*Pleskac, T. J., & Busemeyer, J. R. (2010). Two-Stage Dynamic Signal Detection: A Theory of Choice, Decision Time, and Confidence. Psychological Review, 117, 864-901. doi:10.1037/A0019737*

The model of Pleskac and Busemeyer was developed for the specific situation in which choice and confidence are reported sequentially. Under this circumstance, we (and others) have shown that late information can inform confidence but not choice (e.g., van der Berg et al., *eLife* 2016). This situation, however, does not apply to our tasks. In the human confidence task the choice and the confidence are reported simultaneously. Both were based on short stimulus durations, and subjects used all information to guide their choices (i.e., adding termination bounds did not improve the quality of the fits). This means that there is no “late evidence” that can be used to inform confidence but not choice.

Further, the new analysis of the neurophysiological data from the PDW task (Figure 7) suggests that the information bearing on both choice and confidence are contemporaneous, if not one in the same, as suggested by a recent microstimulation study from our group (Fetsch et al. 2014).